# Progranulin mediates immune evasion of pancreatic ductal adenocarcinoma through regulation of MHCI expression

Phyllis F. Cheung[1,2], JiaJin Yang[1,2], Rui Fang[1,2], Arianna Borgers[1,2], Kirsten Krengel[1,2], Anne Stoffel[1,2], Kristina Althoff[1,2], Chi Wai Yip [3,4], Elaine H. L. Siu[3], Linda W. C. Ng[3], Karl S. Lang[5], Lamin B. Cham[5], Daniel R. Engel[6], Camille Soun[6], Igor Cima [7], Björn Scheffler[7], Jana K. Striefler[8], Marianne Sinn[8], Marcus Bahra[9], Uwe Pelzer [10], Helmut Oettle[11], Peter Markus[12], Esther M. M. Smeets[13], Erik H. J. G. Aarntzen[13], Konstantinos Savvatakis[1,2], Sven-Thorsten Liffers[1,2], Smiths S. Lueong[1,2], Christian Neander[1,2], Anna Bazarna[1,2], Xin Zhang[1,2], Annette Paschen[14], Howard C. Crawford[15,16], Anthony W. H. Chan[17], Siu Tim Cheung [3,18✉] & Jens T. Siveke [1,2✉]

Immune evasion is indispensable for cancer initiation and progression, although its underlying mechanisms in pancreatic ductal adenocarcinoma (PDAC) are not fully known. Here, we characterize the function of tumor-derived PGRN in promoting immune evasion in primary PDAC. Tumor- but not macrophage-derived PGRN is associated with poor overall survival in PDAC. Multiplex immunohistochemistry shows low MHC class I (MHCI) expression and lack of CD8[+] T cell infiltration in PGRN-high tumors. Inhibition of PGRN abrogates autophagy-dependent MHCI degradation and restores MHCI expression on PDAC cells. Antibody-based blockade of PGRN in a PDAC mouse model remarkably decelerates tumor initiation and progression. Notably, tumors expressing LCMV-gp33 as a model antigen are sensitized to gp33-TCR transgenic T cell-mediated cytotoxicity upon PGRN blockade. Overall, our study shows a crucial function of tumor-derived PGRN in regulating immunogenicity of primary PDAC.

[1] Bridge Institute of Experimental Tumor Therapy, West German Cancer Center, University Hospital Essen, Essen, Germany. [2] Division of Solid Tumor Translational Oncology, German Cancer Consortium (DKTK, partner site Essen) and German Cancer Research Center, DKFZ, Heidelberg, Germany. [3] Department of Surgery, Prince of Wales Hospital, The Chinese University of Hong Kong, Hong Kong, China. [4] RIKEN Center for Integrative Medical Sciences, Yokohama, Japan. [5] Institute of Immunology, Medical Faculty, University of Duisburg-Essen, Essen, Germany. [6] Department of Immunodynamics, Institute of Experimental Immunology and Imaging, University Hospital Essen, Essen, Germany. [7] DKFZ-Division Translational Neurooncology at the WTZ, German Cancer Consortium (DKTK partner site Essen/Düsseldorf), Essen, Germany. [8] Universitätsmedizin Charité Berlin, CONKO Study Group, Department of Medical Oncology, Haematology and Tumorimmunology, Berlin, Germany. [9] Department of Surgical Oncology and Robotics, Krankenhaus Waldfriede, Berlin, Germany. [10] Medical Department, Division of Hematology, Oncology and Tumor Immunology, Charité University Hospital, Berlin, Germany. [11] Praxis und Tagesklinik, Dresden, Germany. [12] Department of General, Visceral and Trauma Surgery, Elisabeth Hospital Essen, Essen, Germany. [13] Department of Medical Imaging, Radboud university medical Center, Nijmegen, The Netherlands. [14] Department of Dermatology, University Hospital Essen, University of Duisburg-Essen, Essen, Germany. [15] Rogel Comprehensive Cancer Center, University of Michigan, Ann Arbor, MI, USA. [16] Department of Molecular and Integrative Physiology, University of Michigan, Ann Arbor, MI, USA. [17] Department of Anatomical and Cellular Pathology, Prince of Wales Hospital, The Chinese University of Hong Kong, Hong Kong, China. [18] Li Ka Shing Institute of Health Sciences, The Chinese University of Hong Kong, Hong Kong, China. ✉email: stcheung@surgery.cuhk.edu.hk; j.siveke@dkfz.de

Macrophage-derived progranulin (protein: PGRN; gene: GRN) promoted liver metastasis of pancreatic ductal adenocarcinoma (PDAC)[1] and contributed to immunotherapy resistance in metastatic PDAC[2]. However, the study did not address the role of PGRN in tumor cells. Tumorigenic role of PGRN is well-documented in various cancers, where it promotes cell proliferation, migration, and chemoresistance[3–8]. In HCC, PGRN enhances the shedding of tumor cell MHC class I chain-related protein A (MICA), an innate NK and T-cell stimulatory molecule[9,10], suggesting that PGRN might represent a tumor-intrinsic factor rendering tumor cells invisible to immune elimination. In PDAC, however, the role of tumor-derived PGRN in immune evasion is largely unaddressed.

Although immunotherapy is an established treatment strategy in many cancers, PDAC is so far refractory to immunomodulatory approaches and still remains largely untreatable. The ability of tumor cells to evade immune elimination is fundamental to tumor initiation, progression, and therapy resistance[11,12]. Downregulation of major histocompatibility complex class I (MHCI) is a common mechanism evolved by neoplastic cells to evade immune recognition and cytotoxicity[13,14]. Efficacy of immunotherapies was reported to depend on the expression levels of MHCI on tumor cells[15–17]. Understanding the mechanism to restore tumor MHCI expression is therefore critical to induce antitumor immunity in this deadly disease.

Here, we apply multiplex immunohistochemistry and spatial analysis, as well as functional studies in human PDAC and mouse models of spontaneous PDAC to elucidate the function of PGRN in immune evasion and tumor development of PDAC. Further, we use a spontaneous PDAC mouse model with inducible expression of lymphocytic choriomeningitis virus (LCMV)-glycoprotein (gp)33 as a defined antigen to mechanistically address the effect of PGRN blockade in restoring tumor immunogenicity and tumor antigen-specific cytotoxicity.

## Results

### PGRN exerts distinct functions on tumor cells and macrophages in human PDAC. 
To understand the role of PGRN in human PDAC development, we delineated its expression pattern at different stages in PDAC specimens (Essen cohort, $n = 54$). PGRN expression was observed in preneoplastic cells and remained throughout the malignant transformation (Fig. 1a). The staining was quantified and the number of PGRN$^+$ cells was significantly higher in tumor than non-tumor areas (Supplementary Fig. 1a). Patients were dichotomized into high and low PGRN expression groups (Fig. 1b) with a median of PGRN$^+$ cells as the cutoff. Low PGRN expression showed significantly superior survival than the high expression group (Median overall survival: high vs low PGRN expression group: 9 vs 21mo, Supplementary Fig. 1b). We validated the association of PGRN expression with survival in an independent cohort from Nijmegen ($n = 31$, Supplementary Fig. 1c, d).

Intriguingly, PGRN expression was observed in both tumor and stromal compartments in both Essen (Fig. 1b) and Nijmegen cohorts (Supplementary Fig. 1d). To dissect the roles of PGRN in tumor and stroma separately, we analyzed the RNA-sequencing data set of Maurer et al.[18] (GSE93326, $n = 65$), where PDAC malignant epithelium and stroma were procured by laser capture microdissection. Using Syllogist[19], a reference-based algorithm for cell type estimation, we quantified the signals of 43 different cell types using transcriptomes derived from high GRN ($n = 32$) and low GRN ($n = 32$) stroma samples. In this analysis, macrophages were enriched in high GRN stroma (Supplementary Fig. 1e, Supplementary Data 1). Macrophages also represent the cell type with the highest degree of variable importance in predicting high GRN stroma using random forest analysis (Supplementary Fig. 1f).

Gene set enrichment analysis (GSEA) of the hallmark and KEGG pathways was performed and showed that the pathways enriched for high GRN were different for epithelium and stroma (Fig. 1c, Supplementary Data 2). Notably, in the epithelium but not stroma, high GRN showed downregulation in the allograft rejection gene set, which implies a role in immune evasion; as well as an enrichment in TGF-β signaling, which contributes to immune exclusion and evasion in various cancer types[20]. Besides, no significant correlation was observed between GRN expression in paired epithelium and stroma (Fig. 1d).

We performed multiplex immunofluorescence (mIF) to distinguish PGRN expression derived from tumor cells (PanCK$^+$) and macrophages (CD68$^+$) in tissue microarrays generated from the adjuvant clinical phase III CONKO-001 cohort (gemcitabine vs no treatment, here: trial arm without chemotherapy, $n = 71$)[21,22]. Three different patterns of PGRN expression were observed: PGRN positive signals in (1) both tumor and macrophages; (2) in tumor only; and (3) negative in both cell compartments (Fig. 1e). Survival analysis showed that high PGRN expression in total cells and tumor cells both predicted poor survival, while PGRN expression in macrophages did not predict survival (Fig. 1f). Besides, we analyzed the association of tumor PGRN with various clinicopathological parameters and immune markers that were characterized previously in this cohort[21,22]. Interestingly, high tumor PGRN level was significantly associated with CD8$^+$ cell abundance in negative manner (Supplementary Table 1), implying a potential immunoregulatory role of PGRN expression in primary PDAC.

### PGRN$^+$ tumors of PDAC patients exhibit lower levels of MHCI expression and CD8 infiltration. 
To address the spatial interaction between tumor PGRN and CD8$^+$ cells, we performed mIF in eight cases of human PDAC. Intratumoral heterogeneity was observed in all specimens with high and low PGRN-expressing tumor areas (Fig. 2a). Notably, in high PGRN-expressing tumor regions, we rarely detected CD8$^+$ cells, while the opposite was found in low PGRN-expressing tumor regions of the same tumors (Fig. 2b).

Next, we assessed if tumor PGRN expression was associated with the cytotoxic activity of CD8$^+$ cells, by including the cytotoxic marker granzyme B (GzmB) and MHCI molecule HLA-A in the mIF. In low PGRN-expressing tumor regions, PGRN$^-$/PanCK$^+$ tumor cells showed stronger expression of MHCI molecule HLA-A. Also, higher CD8 infiltration could be observed, in which a significant portion of them was GzmB$^+$ (Fig. 2c). On the contrary, PGRN$^+$/PanCK$^+$ tumor cells expressed no, or only low levels of MHCI, and infiltrating CD8$^+$ and GzmB$^+$ cells were scarce (Fig. 2c). Spatial analysis showed that the percentage of MHCI$^+$ cells in PGRN$^-$/PanCK$^+$ tumor cells was significantly higher than the PGRN$^+$ counterparts. Besides, significantly more CD8$^+$/GzmB$^+$ cells could be found in close proximity (≤50 μM radial distance) of PGRN$^-$/PanCK$^+$ cells (Fig. 2d), implying a potential regulatory role of PGRN in tumor immunogenicity and antitumor cytotoxicity in PDAC.

### PGRN suppression leads to surface MHCI upregulation and dysfunctional autophagy in human PDAC cells. 
To assess the cell-autonomous effect of PGRN on MHCI expression, we examined the surface MHCI molecules HLA-A/B/C, on human PDAC cell lines upon GRN suppression using RNA interference. Among eight human PDAC cell lines, MiaPaCa2 and PaTu8988T cells, which showed the highest PGRN expression

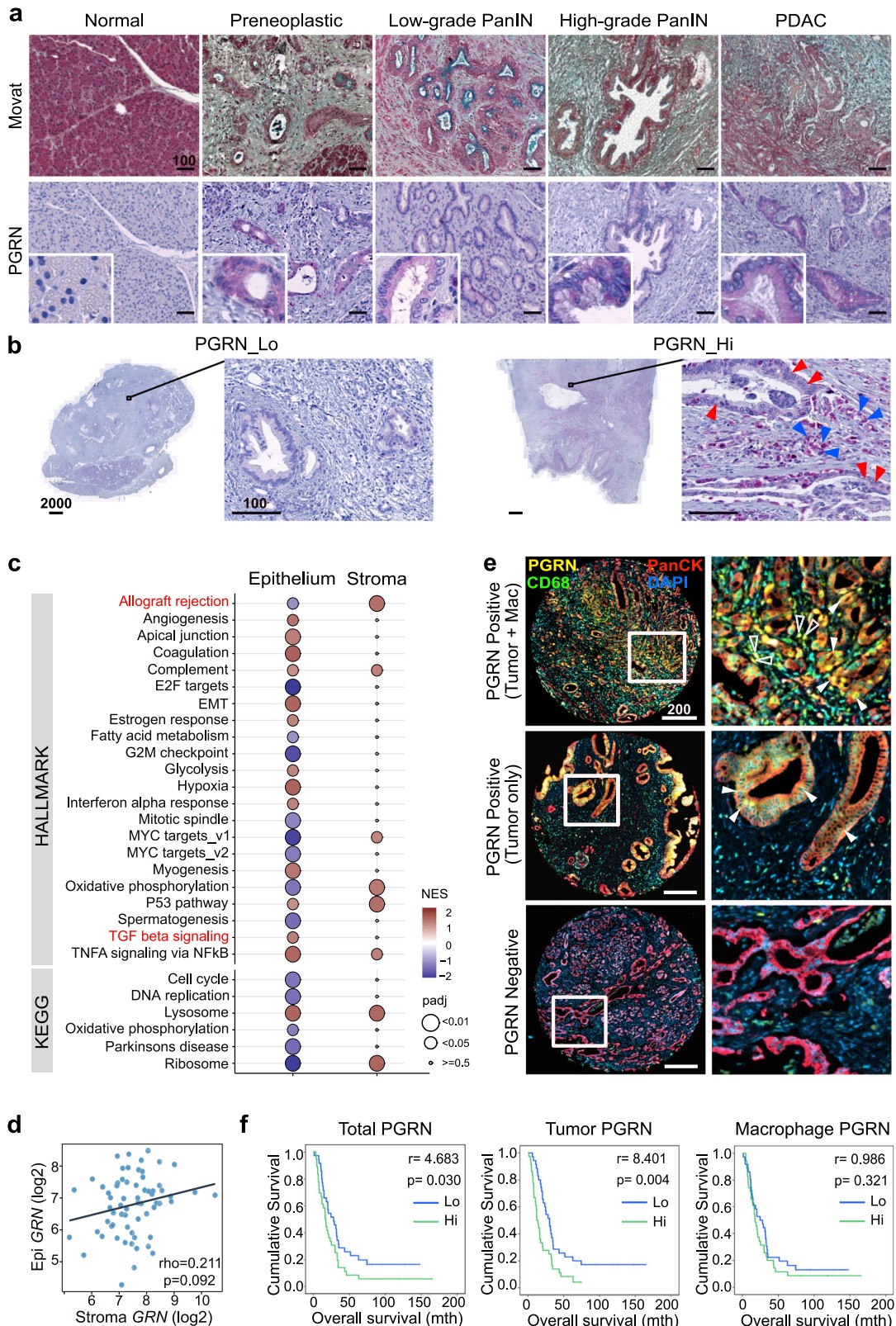

levels (Supplementary Fig. 2a), were selected for *GRN* suppression (Supplementary Fig. 2b, c). Note that MiaPaCa2 and PaTu8988T showed relatively low surface but high intracellular MHCI expression levels (Supplementary Fig. 2d). Surface expression of MHCI on cells of both cell lines was significantly augmented upon RNAi-mediated knockdown of *GRN* (Fig. 3a, Supplementary Fig. 3a). MHCII was also examined and there was no

significant change upon *GRN* suppression in both cell lines (Fig. 3a, Supplementary Fig. 3a), confirming that the PGRN effect was restricted to MHCI. By immunofluorescence (IF), membranous staining of MHCI was observed clearly in *GRN*-suppressed cells, but not in the controls (Fig. 3b, Supplementary Fig. 3b).

In a previous study, pancreatic cancer cells showed down-regulated surface MHCI by autophagy-mediated degradation[23].

**Fig. 1 PGRN exerts different functions on tumor cells and macrophages in human PDAC. a**, **b** Essen cohort ($n = 54$). **a** PGRN expression pattern during human PDAC development. Movat and IHC staining of PGRN in normal pancreas, preneoplastic low- and high-grade PanIN stages, and tumorous tissues of PDAC patients. ($n = 54$). Representative images are shown. **b** IHC staining of the representative patient specimens with high and low PGRN levels as defined by the quantification of Definiens, with a median of a number of PGRN$^+$ cells as cutoff. Right panel: PGRN expression in both tumor and stromal compartments of PDAC. Red arrowheads indicate PGRN$^+$ tumor cells; blue arrowheads indicate PGRN$^+$ stromal cells. ($n = 54$). Representative images are shown. **c**, **d** Maurer et al. data set (GSE93326, $n = 65$). **c** GSEA result of hallmark and KEGG pathways in PDAC epithelium and stroma samples shows different enriched gene sets for high *GRN* in epithelium and stroma. Enrichment was tested against the differential expression profile of *GRN*-high ($n = 32$) versus *GRN*-low ($n = 32$) in epithelium and stroma samples separately. Pathways with the Benjamini–Hochberg method adjusted $p$ value (padj) smaller than 0.05 were considered significant. Pathways with padj < 0.01 from either epithelium or stroma groups were shown with their NES and padj values. A complete list of significant pathways is shown in Supplementary Data 2. Gene sets of interest are highlighted in red. **d** No significant correlation between epithelium (tumor) and stroma *GRN* expression levels (Log2CPM) in 63 pairs of PDAC specimens. **e**, **f** CONKO-001 cohort ($n = 71$). Spearman's rank correlation: rho = 0.211, $p = 0.092$. Tissue microarrays were stained for PGRN, CD68 (macrophage), PanCK (tumor), and DAPI by multiplex immunofluorescent (mIF) staining and quantified by Definiens. **e** Representative tissue cores show PGRN expression in: both tumor cells (Tumor) and macrophages (Mac), in tumor cells only, and negative in both cell types. ($n = 71$). Representative images are shown Right column: filled arrowheads indicate PGRN$^+$ tumor cells. Hollow arrowheads indicate PGRN$^+$ macrophages. **f** PDAC samples were categorized into high ($n = 35$) and low ($n = 36$) expression groups (cutoff: median of the number of PGRN$^+$ cells) based on the PGRN expression in all the cells (Total PGRN), tumor cells (Tumor PGRN, PGRN$^+$PanCK$^+$) only, and macrophages (macrophage PGRN, PGRN$^+$CD68$^+$) only. Kaplan–Meier overall survival plots according to PGRN expression level in different cell compartments. Log-rank test. Scale bar unit: μm.

Coincidentally, PGRN was reported to regulate autophagy in various cell types[24,25]. PGRN-deficient mice showed defective degradation of autophagosomes under starvation[26–28]. Therefore, we reasoned that PGRN might control the surface expression of MHCI through autophagy. Firstly, we co-stained MHCI and autophagosome marker LC3B in the cells. Upon *GRN* suppression, the number and size of LC3B puncta increased significantly when compared to the controls (nc) (Fig. 3c, d, Supplementary Fig. 3c, d), indicating an increase in autophagosomes in *GRN*-suppressed cells. However, autophagosomes can be increased by two different mechanisms: increased autophagosome synthesis or impaired clearance of autophagosomes. Therefore, we assessed LC3B-II levels in the cells treated with or without V-ATPase inhibitor Balfinomycin A (BafA), which blocks the degradation of autophagosomes. If autophagy is induced upon *GRN* suppression, BafA treatment will increase the LC3B-II level; if clearance of autophagosomes is blocked, LC3B-II level will not be affected in the presence of BafA. Here, BafA increased LC3B-II in control cells (nc), but not the *GRN*-suppressed PDAC cells, which already showed augmented LC3B-II level when compared to controls (Fig. 3e, S3e). The phenomenon was further confirmed by measuring the amount of p62/sequestosome-1(SQSTM1), which is involved in autophagic cargo recognition and is lost in the late stage of autophagy during autolysosome degradation[29]. An increase in the amount of p62/SQSTM1 indicates inhibition of autophagic flux. Immunoblotting results showed that p62/SQSTM1 levels increased in *GRN*-suppressed cells (Fig. 3e, Supplementary Fig. 3e). This indicates that *GRN* suppression induces autophagosome accumulation, reflecting inhibition of their degradation in PDAC cells. As expected, the increase in p62 was also observed in cells treated with BafA when compared to nc. However, no prominent change was observed for *GRN*-suppressed cells (Fig. 3e, Supplementary Fig. 3e).

*GRN* suppression inhibited the degradation stage of autophagy indicated it might be involved in the function of the lysosome. Indeed, PGRN is reported to regulate lysosomal functions and biogenesis through lysosomal acidification[24]. We measured lysosome by Cytopainter lysogreen indicator in the *GRN*-suppressed cell lines by flow cytometry, and found that accumulation of lysosomes was significantly increased in sh*GRN* transfectants (Fig. 3f, Supplementary Fig. 3f).

Next, we examined the subcellular localization of PGRN and the effects of *GRN* suppression on the autophagy pathway. IF staining of PGRN, lysosome marker Lamp1 and late endosome marker Rab7 showed that, PGRN co-localized with Rab7$^+$

vesicles and partially overlapped with Lamp1$^+$ vesicles (Fig. 3g, Supplementary Fig. 3g). Upon *GRN* suppression, the staining intensity of Lamp1$^+$ vesicles increased significantly, while that of Rab7$^+$ vesicles was not changed (Fig. 3g, Supplementary Fig. 3g). Next, we examined the effects of PGRN on lysosomal function. Because acidic pH is required for lysosomal activity[30], we evaluated whether PGRN affected lysosomal pH using LysoSensor DND-189, which emits a stronger fluorescent signal in an acidic environment. *GRN* suppression induced a significant reduction in DND-189 signal when compared to control cells (Fig. 3h, Supplementary Fig. 3h), indicating an important role of PGRN in maintaining lysosomal acidification. Next, *GRN*-suppressed cells were assayed for their ability to process DQ-BSA (a derivative of BSA), which is dequenched upon cleavage by proteolytic enzymes in lysosomes[31]. As shown by IF staining and flow cytometry, dequenching of DQ-BSA was significantly decreased in *GRN*-suppressed cells when compared to nc controls (Fig. 3i, j, Supplementary Fig. 3i, j). Upon treatment with BafA, dequenching of DQ-BSA was greatly diminished and there was no difference between *GRN*-suppressed cells and controls (Fig. 3i, j, Supplementary Fig. 3i, j). The above findings indicate the crucial role of PGRN in regulating proteolytic activity.

Finally, to further confirm the effect of PGRN, we treated a low PGRN-expressing PDAC cell line with high surface MHCI expression, HupT4 (Supplementary Fig. 2a, d), with recombinant PGRN (rPGRN). Upon treatment with rPGRN, the surface MHCI level of HupT4 cells decreased (Supplementary Fig. 4a). IF staining showed that the clear membranous MHCI staining in HupT4 cells was greatly reduced upon rPGRN treatment (Supplementary Fig. 4b). LC3B puncta were increased upon rPGRN treatment (Supplementary Fig. 4b, c). LC3B-II level in the cells was assessed by immunoblotting. Notably, rPGRN increased LC3B-II level in HupT4 cells (Supplementary Fig. 4d), which was further augmented upon co-treatment with BafA (Supplementary Fig. 4e). Taken together, our data demonstrated an important role of PGRN in lysosomal activity and autophagic flux.

**PGRN blockade suppresses tumorigenesis in vivo**. Next, we investigated the effect of PGRN inhibition on PDAC development in vivo using an antibody (Ab)-based PGRN blockade strategy. Anti-PGRN Ab was previously shown to neutralize soluble PGRN (sPGRN) in liver cancer[9,32,33]. Since PGRN is a secreted glycoprotein regulating its own expression in autocrine/paracrine manner, neutralization of sPGRN reduces both extracellular and intracellular PGRN[32]. We treated MiaPaCa2 and

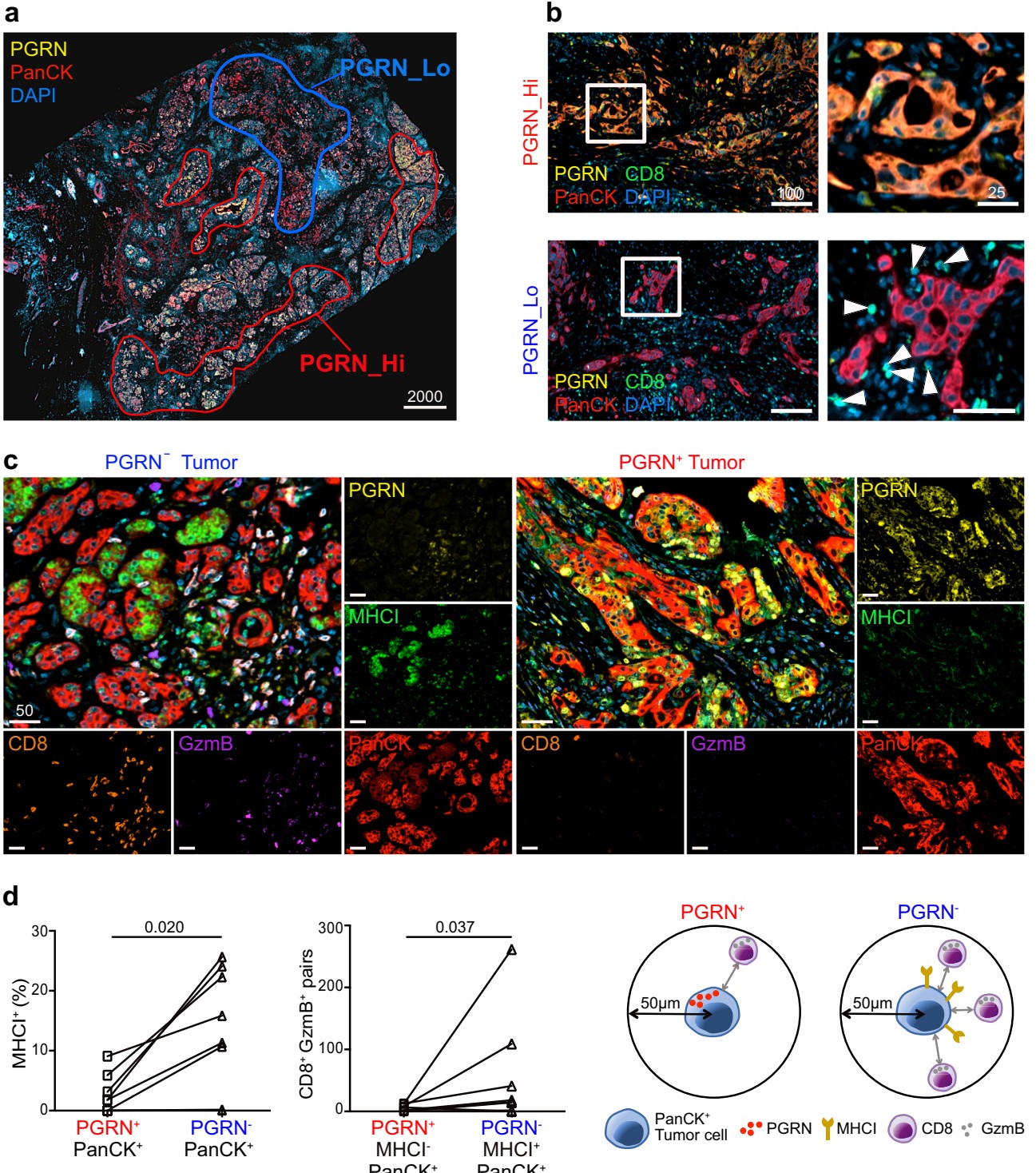

**Fig. 2 PGRN$^+$ tumors of PDAC patients exhibit lower levels of MHCI expression and CD8 infiltration. a** Representative mIF image demonstrates the intratumoral heterogeneity of PGRN expression in human PDAC, where high and low PGRN-expressing tumor regions were observed in the same specimen. ($n = 8$). Representative images are shown. **b** mIF staining of PGRN (yellow), CD8 (green), and PanCK (red) in human PDAC shows increased CD8 infiltration in low PGRN-expressing tumor areas. ($n = 8$). Representative images are shown. **c** Differential MHCI (HLA-A) expression in PGRN$^+$ and PGRN$^-$ tumors, and the infiltration of GzmB$^+$CD8$^+$ cells in their corresponding neighborhoods. ($n = 8$). Representative images are shown. **d** Automated computational analysis showing the percentage of MHCI$^+$ cells in PGRN$^+$PanCK$^+$ and PGRN$^-$PanCK$^+$ tumor populations, and the number of CD8$^-$GzmB$^-$ cells in proximity (<50 μM radical distance) of PGRN$^+$MHCI$^-$ or PGRN$^-$MHCI$^+$PanCK$^+$ tumor cells in human PDAC ($n = 8$). Two-tailed Mann–Whitney test. Scale bar unit: μm.

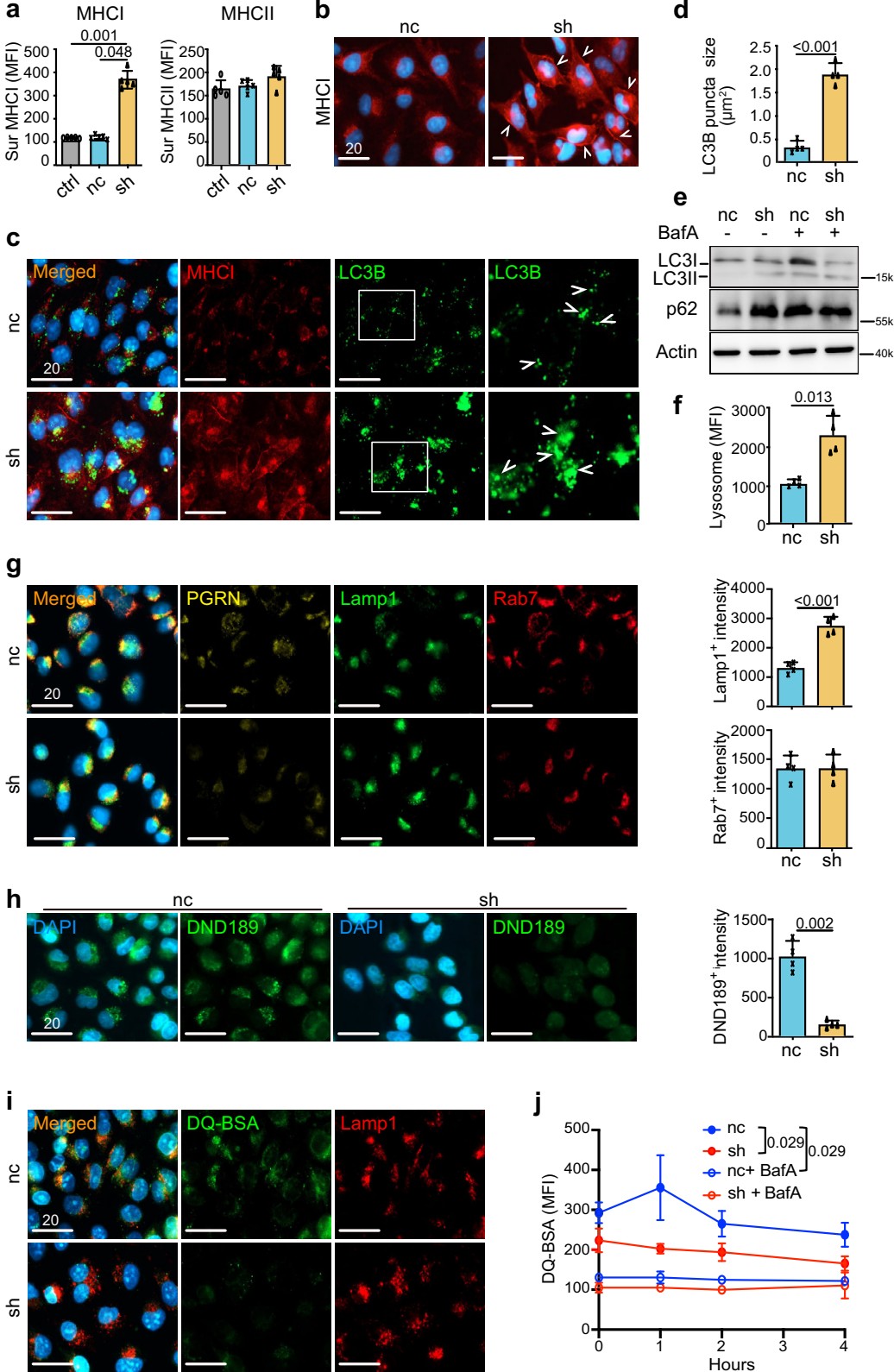

PaTu8988T cells with anti-PGRN Ab in vitro, and both cellular and secretory PGRN levels were significantly reduced upon Ab treatment (Supplementary Fig. 5a, b), supporting subsequent targeting of cellular and circulating PGRN in vivo.

We delineated PGRN expression pattern during PDAC development in a well-characterized spontaneous endogenous PDAC mouse model (termed CKP) with conditional oncogenic Kras[G12D] mutation and loss of Tp53[34,35], which allows longitudinal characterization of tumor evolution. Pancreata were harvested at the stages preneoplasia (4 weeks), early (6 weeks), and advanced PDAC (>8 weeks). mIF showed tumor PGRN (PGRN[+]PanCK[+]) expression in preneoplastic lesions

**Fig. 3 PGRN suppression leads to surface MHCI upregulation and dysfunctional autophagy in human PDAC cells. a** Surface MHCI (HLA-A/B/C) and MHCII (HLA-DR) expression on human PDAC cell line MiaPaCa2 upon *GRN* suppression was assessed by flow cytometry (n = 5 independent experiments). One-way ANOVA, Kruskal–Wallis test. **b** IF staining of MHCI marker HLA-A/B/C demonstrates augmented surface MHCI expression on MiaPaCa2 upon *GRN* suppression. White arrowheads indicate the membraneous staining of MHCI. (n = 4 independent experiments). Representative images are shown. **c, d** IF staining of MHCI (red) and LC3B (green) in MiaPaCa2 cells upon *GRN* suppression. **d** The average size of LC3B puncta of 50 cells of each treatment was measured by ZEN software. (n = 4 independent experiments). Two-tailed Mann–Whitney test. **e** Western blot showing LC3B (LC3I, II), p62/SQSTM1 (p62), and actin of MiaPaCa2 cells upon *GRN* suppression treated with or without V-ATPase inhibitor Balfinomycin A (BafA, 100 nM, 24 h). (n = 4 independent experiments). Representative images are shown. **f** Lysosome content in MiaPaCa2 upon *GRN* suppression was assessed by staining with Cytopainter LysoGreen indicator and measured by flow cytometry. n = 4 independent experiments. Two-tailed Mann–Whitney test. **g** IF staining of PGRN, lysosome marker Lamp1, and late endosome marker Rab7 in MiaPaCa2 upon *GRN* suppression. Right panel: average intensity of Lamp1 and Rab7 signals per cell was measured by HALO software. n = 4 independent experiments. Representative images are shown. Two-tailed Mann–Whitney test. p = 0.029 (Lamp1[+]). **h** IF staining of LysoSensor DND-189 in MiaPaCa2 cells upon *GRN* suppression. Right panel: average intensity of DND-189 signal per cell was measured by HALO software. n = 4 independent experiments. Two-tailed Mann–Whitney test. **i** IF images showing the dequenched DA-BSA (green) and Lamp1 (red) in MiaPaCa2 cells upon *GRN* suppression. **j** Quantification of the dequenched DQ-BSA signal in MiaPaCa2 cells upon *GRN* suppression, with or without prior treatment with BafA (100 nM, 24 h), after different chasing times. n = 4 independent experiments. Two-tailed Mann–Whitney test. ctrl: parental PDAC cells; nc: shRNA scrambled control; sh; *GRN* shRNA. *MFI* mean fluorescence intensity. Mean + SD are shown. Scale bar unit: μm.

recapitulating expression patterns in human PDAC. As lesions progressed to early PDAC, tumor PGRN levels reached the maximum, and slightly decreased in the advanced stage (Fig. 4a).

Anti-PGRN Ab treatment started when mice were 4 weeks old (preneoplasia) and was terminated at 6 weeks, when PDAC formation is frequently observed in this model (Fig. 4b). Anti-PGRN Ab treatment significantly alleviated tumor burden as compared with controls in terms of tumor weight (Fig. 4c, d) and proportion of tumorous (PanCK+) tissues (Fig. 4e, f). IHC analysis confirmed that PGRN+ cells in Ab-treated tumors were significantly reduced (Fig. 4g). Notably, Ki67+ tumor cells were also significantly reduced, indicating reduced tumor proliferation (Supplementary Fig. 5c). Thus, PGRN blockade prominently halts tumor initiation and progression in this aggressive genetic model of spontaneous PDAC development.

**In vivo PGRN blockade suppresses M2 polarization but not fibroblast accumulation.** Macrophage-derived PGRN associates with M2 phenotypes[1,2]. Here, we examined the effect of PGRN blockade on TAM infiltration and fibrosis. Pan-macrophage marker F4/80 level was not significantly changed upon PGRN blockade (Fig. 4h). However, TAM marker MRC1 level was greatly reduced, while M1 markers phospho-STAT1 and iNOS increased upon treatment (Fig. 4h), indicating that PGRN blockade skewed macrophage polarization from M2 to M1 phenotype.

Our results regarding the effect of PGRN on M2 polarization echo the previous findings by Nielsen et al.[2] in metastatic PDAC, in which macrophage-derived PGRN-induced fibrosis in a non-cell-autonomous manner and blocked CD8 infiltration. Therefore, we also examined the abundance of fibroblasts in the PGRN Ab-treated tumors. Paradoxically, no significant difference was observed in fibroblast accumulation (α-SMA+ cells) in the tumors (Supplementary Fig. 5d), suggesting that the revived CD8 infiltration and activation induced by PGRN blockade is not mediated by fibrosis reduction. Given the heterogeneity of cancer-associated fibroblasts (CAFs), in addition to α-SMA+ myofibro-blasts (myCAFs), we also examined the effect of PGRN on another important CAF subtype, the inflammatory CAFs (iCAFs). By co-staining podoplanin (PDPN) and Ly6C using mIF, iCAFs (PDPN+Ly6C+) were identified and were found to be reduced in *CKP* tumors treated with PGRN Ab, although statistical significance is not reached (Supplementary Fig. 5e). By flow cytometry, we assessed the abundance of various CAF subtypes, including the relatively rare antigen-presenting CAFs (apCAFs), based on co-staining of PDPN, Ly6C, and MHCII[36]

(Supplementary Fig. 5f). Here, we observed a slight decrease again in iCAFs upon PGRN Ab (Supplementary Fig. 5g), although statistical significance is not reached. However, PGRN Ab, in general, did not prominently change the composition as well as an abundance of various CAFs (Supplementary Fig. 5g).

**In vivo PGRN blockade revives CD8 antitumor cytotoxicity.** Next, we focused on the effect of PGRN blockade on the tumor immune microenvironment. We found the number of infiltrating CD3+ and CD8+ cells significantly increased upon Ab treatment (Fig. 5a). The increase of CD8+ cells was not solely due to the delayed tumor progression. The dynamic of immune landscape throughout PDAC development was previously delineated in this model[34] and CD8+ cells contributed to less than 4% of cells in tumors (re-quantified in whole-tissue scale, Supplementary Fig. 6a). Upon Ab treatment, we observed up to 10% CD8+ cells in the tumors (Fig. 5a), indicating that PGRN blockade promotes additional CD8+ cell infiltration. Importantly, the expression of cytotoxic markers granzyme B, T-bet, and Eomes was also significantly augmented (Fig. 5a, S6b). mIF confirmed that a significant portion of infiltrating CD8+ cells was also positive for cytotoxic markers granzyme B or T-bet (Fig. 5b), suggesting PGRN Ab restored the infiltration of CD8 cells that were cytotoxically active. The antitumor cytotoxicity induced by PGRN blockade was confirmed by an increased level of cleaved caspase 3 in tumor cells (Fig. 5c). The number of foxp3+ cells was significantly suppressed upon PGRN blockade (Supplementary Fig. 6b). However, since the foxp3 expression level was low in the *CKP* model, we consider it unlikely that the revived CD8 cytotoxicity was caused by the suppression of regulatory T cells. Finally, CD4 infiltration remained unchanged upon PGRN blockade (Supplementary Fig. 6b), indicating that the increase in M1 macrophages (Fig. 4h) might not be sufficient to attract the CD4+ T cells.

To demonstrate that CD8 cells are the main effectors of antitumor cytotoxicity upon PGRN blockade, we performed the in vivo PGRN blockade in *CKP* mice with systemic CD8 depletion using an Ab (Fig. 5d). Upon CD8 depletion, the reduction of tumor burden induced by PGRN Ab was greatly abolished (Fig. 5e, f). Depletion of CD8 cells was confirmed by flow cytometry (Fig. 5g). IHC analysis showed that the PGRN Ab-induced reduction of PanCK+ tissues was restored by co-treatment of CD8 depletion Ab, while increased tumor infiltration of CD8+ and granzyme B+ cells were prominently abrogated (Fig. 5h). As expected, PGRN Ab-induced cytotoxicity as reflected by cleaved caspase 3 staining in the tumors was also not observed

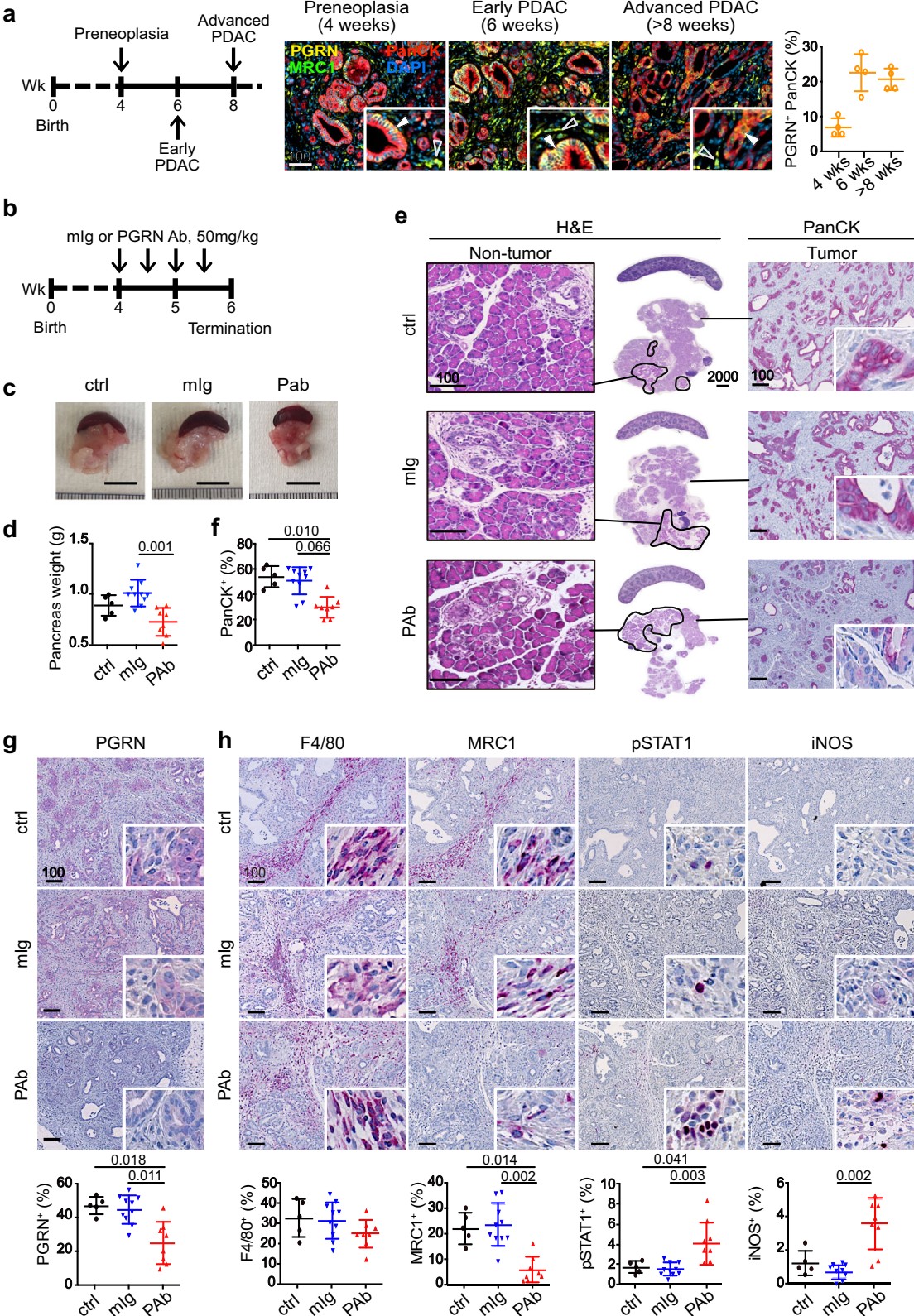

upon CD8 depletion (Fig. 5i). All the above findings confirm the antitumor cytotoxicity induced by PGRN blockade is mediated largely by CD8 cells.

**In vivo PGRN blockade restores tumor MHCI expression that is spatially associated with increased CD8 cell infiltration.** Next, we assessed the expression of MHCI molecule H-2Db in the *CKP*

tumors. H-2Db expression, which appeared low or absent in tumor cells, was significantly restored upon PGRN Ab treatment (Fig. 6a). This finding was validated by flow cytometric analysis of freshly dissociated tumor cells from the *CKP* tumors. In addition to H-2Db, another MHCI molecule H-2Kb, was also significantly increased in tumors treated with PGRN Ab (Supplementary Fig. 7a). The effect of PGRN Ab on surface MHCI expression was

**Fig. 4 In vivo PGRN blockade suppresses tumor initiation and progression in mouse PDAC. a** mIF staining and quantification of PGRN+ tumor cells (PGRN+PanCK+) in the pancreas of wild-type mice and *CKP* mice with preneoplasia (4 weeks), early (6 weeks) and advanced (>8 weeks) PDAC. Filled arrowheads indicate PGRN+ tumor cells; Hollow arrowheads indicate PGRN+ macrophages. **b** Timeline for treatment of *CKP* mice with mouse isotype (mIg) or anti-PGRN antibody (PAb), 50 mg/kg. **c** Representative pictures of tumors and spleens from mIg-treated (mIg, n = 10), PGRN Ab-treated (PAb, n = 8), and untreated (ctrl, n = 5) *CKP* mice. **d** Weight of pancreas of *CKP* mice upon dissection. One-way ANOVA, Kruskal–Wallis test. **e** PGRN blockade suppresses tumorigenesis in *CKP* mice. Left panel: H&E staining of pancreas and spleens of *CKP* mice. Non-tumorous pancreatic tissues are highlighted by black lines. Right panel: IHC staining of panCK in the *CKP* pancreata. **f** The percentage of PanCK+ cells in the whole pancreata was quantified by Definiens software. One-way ANOVA, Kruskal–Wallis test. IHC staining of (**g**) PGRN and (**h**) pan-macrophage marker F4/80, M2 marker MRC1, M1 markers pSTAT1 and iNOS in *CKP* tumors treated with or without PGRN Ab (PAb) or mIg (50 mg/kg). The lower panels show the percentage of respective positive cells in the whole tumorous tissues as quantified by Definiens software. ctrl: n = 5; mIg: n = 10; PAb: n = 8. Mean + or ± SD is shown. One-way ANOVA, Kruskal–Wallis test. Scale bar unit: μm.

also observed on human PDAC cell lines (Supplementary Fig. 7b). However, MHCII expression was not changed upon PGRN treatment (Fig. 6a), which echoes earlier findings on the unchanged CD4 infiltration (Supplementary Fig. 6b). We also examined lysosome marker LAMP1 and autophagosome marker LC3B in PGRN Ab-treated *CKP* tumors. Both markers were significantly increased upon PGRN blockade (Supplementary Fig. 7c), implying the presence of dysfunctional autophagy that might be involved in the PGRN blockade-mediated MHCI expression.

mIF was performed to illustrate the spatial association among PGRN, MHCI, and CD8 cells. We observed heterogeneous expression patterns of PGRN and MHCI across the PGRN Ab-treated tumor (n = 8, Fig. 6b). Notably, MHCI expression was absent in the PGRN+ region with advanced PDAC tumor, while it was observed in the PGRN− region with mostly preneoplastic lesions (Fig. 6b). The spatial analysis demonstrated that the proportion of MHCI+ cells was significantly higher in PGRN− tumor cells (Supplementary Fig. 7d). Besides, significantly more CD8 + cells were found in the vicinity (≤50 μm radical distance) of PGRN−MHCI+ tumor cells when compared to PGRN+ counterparts (Fig. 6c, S7d).

**In vitro PGRN blockade promotes CD8 antitumor cytotoxicity via MHCI regulation**. Next, we investigated if PGRN-induced MHCI restoration could effectively lead to functional effects on antitumor cytotoxicity. Due to the low mutational burden, neoantigen levels in PDAC are relatively low. Thus, we generated a dual recombination (cre/lox;flp/frt) next-generation mouse model with spatial and temporal controlled antigen expression, namely gp33 of LCMV, in tumor cells. Here, flp-mediated activation of KrasG12D and loss of Tp53 in pancreatic progenitors (*FKP* model) leads to spontaneous PDAC development mirroring tumorigenesis in the cre-mediated *CKP* model. We interbred *FKP* mice with Rosa26-LSL-GP and Rosa26-FSF-CreERT2 mice (*FKPC2GP*), in which CreERT2 expressing pancreatic cells express LCMV-gp33 upon tamoxifen administration (Fig. 7a). We established a primary pancreatic cancer cell line, GP82, from an *FKPC2GP* mouse and induced LCMV-gp33 expression in GP82 in vitro. The induced GP82 cells were co-cultured with LCMV-gp33-reactive T cells isolated from the spleen of P14-TCR-Tg mice (Fig. 7b), to assess the effect of PGRN Ab on tumor-specific cytotoxicity.

We first confirmed LCMV-gp33 expression in GP82 cells upon tamoxifen treatment (Fig. 7c) and treated them with PGRN Ab or mIg for 2 consecutive days. PGRN was significantly down-regulated upon Ab treatment (Fig. 7d). Congruent with findings in *GRN*-suppressed human cell lines, the surface expression of MHCI molecule H-2Db and H-2Kb on GP82 cells increased significantly by PGRN Ab treatment (Fig. 7e, Supplementary Fig. 8a). IF staining also demonstrated that H-2Db expression in

PGRN Ab-treated cells was greatly increased with membranous localization (Fig. 7f).

Upon co-culture, LCMV-gp33-reactive T cells were found to accumulate close to GP82 cells induced for LCMV-gp33 expression, but not to those without induction (Fig. 7g, S8b, c). Upon treatment with PGRN Ab, the accumulation of T cells in the vicinity of tumor cells was markedly potentiated. However, the T-cell accumulation was prominently abrogated upon MHCI blockade with H-2Db-neutralizing Ab (Fig. 7g, Supplementary Fig. 8b, c). Importantly, the cytotoxicity level of LCMV-gp33-induced cells upon PGRN Ab treatment was significantly increased as compared with control, and the PGRN Ab-induced cytotoxicity was abrogated upon addition of anti-MHCI neutralizing Ab (Fig. 7h, S8d). We next assessed T-cell cytotoxic activity in the co-culture system. Quantities of granzyme B+, TNF+, and IFNg+ CD8+ cells were all significantly increased in PGRN Ab treatment when compared to the controls (Fig. 7i, S8e), which again, was significantly diminished with the anti-MHCI neutralizing Ab, confirming the indispensable role of MHCI in the PGRN Ab-induced tumor-specific cytotoxic effect.

**In vivo PGRN blockade promotes antigen-specific T-cell cytotoxicity against tumors**. Next, we validated the above effect of PGRN Ab treatment in vivo. Fig. 8a illustrates the experimental setup and treatment timeline. GP82 cells were transplanted orthotopically into C57BL/6 J mice. Upon tumor formation confirmation by ultrasound imaging, tamoxifen was injected to induce LCMV-gp33 expression in the tumor. After two tamoxifen injections, PGRN Ab (n = 4) or corresponding isotype control (mIg, n = 4) was given twice a week for 2 weeks. LCMV-gp33-reactive T cells, freshly isolated from spleens of P14-TCR-Tg mice, were injected intravenously one day after the first administration of PGRN Ab or mIg. Mice were terminated one day after the last administration of Ab treatment.

LCMV-gp33 expression in the tumors was confirmed, using the GP82 xenografts that were not treated with tamoxifen as controls (Supplementary Fig. 9a, b). Strikingly, tumor size was prominently reduced upon PGRN Ab treatment when compared to mIg control (Fig. 8b). The progression of tumor volume was monitored by ultrasound imaging. Tumor increase in PGRN Ab-treated tumor was slightly lower than that of mIg treatment, though not statistically significant due to limited subject numbers (Fig. 8c). Importantly, however, total T cell (CD3+) and exogenous LCMV-gp33-reactive T-cell (CD45.1+) infiltration were both increased upon PGRN Ab treatment (Fig. 8d, Supplementary Fig. 9c). Furthermore, levels of GzmB, TNF, and IFNg were all significantly increased in PGRN Ab-treated tumors (Fig. 8e, Supplementary Fig. 9c). IHC confirmed that in PGRN Ab-treated mice, PGRN levels were reduced while MHCI, CD8+ T-cell infiltration, GzmB, and cleaved caspase 3 levels were all substantially increased compared to controls (Fig. 8f). As expected, in control mice without LCMV-gp33 expression and

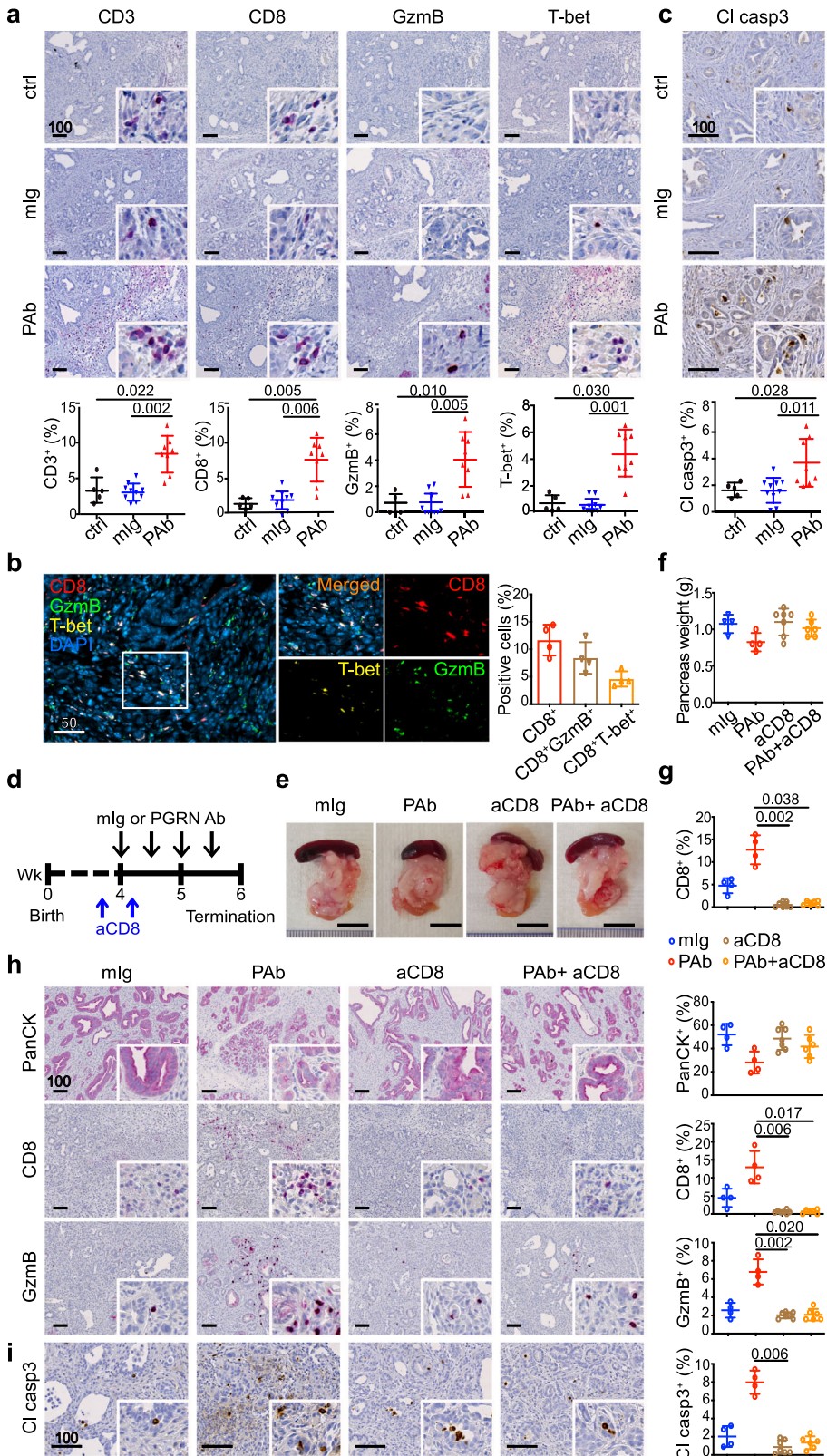

PGRN Ab, in which MHCI expression was not increased, injection of LCMV-gp33-reactive T cells did not elicit any CD8 infiltration and cytotoxic activity (Supplementary Fig. 9d, f). Notably, another control group with both LCMV-gp33 expression and PGRN Ab treatment, but without LCMV-gp33-reactive T-cell injection, showed upregulated MHCI level, and CD8 infiltration and antitumor cytotoxicity was also increased

(Supplementary Fig. 9e, f). However, the magnitude of the increase, particularly GzmB and cleaved caspase 3 levels, were smaller than those with LCMV-gp33-reactive T cells (Supplementary Fig. 9e, f). Besides, no significant benefit in tumor growth suppression could be observed (Supplementary Fig. 9g). The results indicate the importance of the interaction between tumor-specific antigen and antigen-reactive T cells in eliciting

**Fig. 5 In vivo PGRN blockade revives CD8 antitumor cytotoxicity.** IHC staining of **a** T-cell markers CD3 and CD8; cytotoxic markers granzyme B (GzmB), and T-bet in *CKP* tumors treated with or without PGRN Ab (PAb) or mIg (50 mg/kg). The lower panels show the percentages of positive cells in the whole tumor. ctrl: $n = 5$; mIg: $n = 10$; PAb: $n = 8$. One-way ANOVA, Kruskal–Wallis test. **b** mIF showing the proportion of $CD8^+$ cells co-expressing cytotoxic markers GzmB or T-bet. $n = 4$. **c** IHC staining of apoptotic marker cleaved caspase 3 (Cl casp3) in *CKP* tumors treated with or without PGRN Ab (PAb) or mIg. The lower panels show the percentages of positive cells in the whole tumor. ctrl: $n = 5$; mIg: $n = 10$; PAb: $n = 8$. One-way ANOVA, Kruskal–Wallis test. **d** Timeline for treatment of *CKP* mice with CD8 depleting Ab (aCD8, 25 mg/kg), mouse isotype (mIg) or anti-PGRN antibody (PAb, 50 mg/kg). **e** Representative pictures of tumors and spleens from *CKP* mice. **f** Weight of pancreas of *CKP* mice upon dissection. mIg: $n = 4$; PAb: $n = 7$; aCD8: $n = 7$; aCD8+ PAb: $n = 6$. **g** Quantification of tumor-infiltrating CD8 in the *CKP* tumors by flow cytometry. One-way ANOVA, Kruskal–Wallis test. IHC staining of **h** PanCK, CD8, GzmB, and **i** cleaved casp3 in *CKP* tumors treated with or without PGRN Ab, mIg and/or CD8 depleting Ab (aCD8). mIg: $n = 4$; PAb: $n = 4$; aCD8: $n = 7$; aCD8+ PAb: $n = 6$. One-way ANOVA, Kruskal–-Wallis test. Mean ± SD is shown. Scale bar unit: μm.

profound antitumor cytotoxicity. Besides, since functional macrophages are present, the results also imply that the effect of PGRN Ab on macrophages as shown earlier does not contribute prominently to the antitumor cytotoxicity in this model system. Overall, we conclude that PGRN blockade promotes antigen-specific T-cell cytotoxicity in endogenous PDAC.

## Discussion

Understanding the mechanisms exploited by the tumors is critical to overcoming immune evasion. PDAC has relatively few coding mutations, and thus few neoantigenic targets. In light of this, antigenic tumor peptides or dendritic cells loaded with shared peptides have been recently introduced to the clinic[27,37]. However, despite the activation of specific antitumor T-cell immunity, the observed tumor regressions are so far below expectations, and one of the possible reasons could be the absence or low level of tumor MHCI expression. Here, we showed an imperative role of PGRN in tumor cells as a key instructive regulator of immune evasion via MHCI regulation in PDAC cells. We provide evidence that PGRN blockade restored MHCI expression by inhibiting lysosomal activity and degradation of autophagosomes, which concurs with the previous study showing the role of autophagy in promoting immune evasion by MHCI degradation in pancreatic cancer[23]. Indeed, mounting evidence has demonstrated the tumor-promoting role of autophagy by inhibiting T-cell immune responses[38,39]. In addition to MHCI regulation, autophagy also limits T-cell cytotoxicity by interfering with the interferons and TNF pathways[38–40]. Notably, autophagy inactivation in tumor cells was shown to enhance the efficacy of immune checkpoint blockade in mouse models[40–42], further supporting the therapeutic potential of targeting autophagy to enhance antitumor immunity.

In fact, in addition to autophagy, there are various mechanisms regulating MHCI expression in tumor cells. A previous study demonstrated that KRAS drove immune evasion in PDAC and identified MYC and BRAF as key mediators[43]. Profound changes including increased tumor infiltration of T cells and upregulated MHC gene expression were observed in KRAS KO tumors. Interestingly, MYC and SMAD4 played opposing roles in PDAC maintenance, and the authors proposed a possible role of the TGF-β/smad4 signaling pathway as an escape mechanism from oncogenic KRAS addiction in PDAC development. While our preliminary analyses in the CONKO-001 patient cohort, albeit at a low sample number, did not show a significant association between PGRN expression and KRAS mutation status (Supplementary Table 2) and cell lines showed no differences in MYC expression upon PGRN modulation (Supplementary Fig. 10a), a significant reduction of TGF-β expression in *GRN*-suppressed PDAC cells was observed (Supplementary Fig. 10b). In addition, a TGF-β gene signature was significantly enriched in *GRN*-high tumors of the Maurer et al.[18] dataset (Fig. 1c), implying a potential link between PGRN-induced immunomodulation and

TGF-β signaling. However, given the complexity of the TGF-β pathway, further in-depth research is required to assess the interaction among PGRN, TGF-β, and KRAS signaling in regulating immune evasion in PDAC.

PGRN blockade with Ab was used in vivo to demonstrate the functional effect of PGRN. Since PGRN is expressed in both tumor cells and macrophages, where different biological effects are exerted, our Ab approach is not able to distinguish the functions and significance of PGRN in the two compartments. However, from a translational perspective, neutralizing the secretory PGRN and subsequently reducing cellular PGRN to achieve systemic PGRN blockage by PGRN Ab treatment is of value for future therapeutic development. Indeed, in addition to MHCI regulation, we anticipate that PGRN also plays a crucial role in other immune evasion mechanisms. In our study, we showed that PGRN blockade induced macrophage polarization from M2 to M1 phenotype (Fig. 4h), which echoes the previous findings reported in metastatic PDAC in liver[1]. The finding suggests that PGRN blockade not only induces CD8-mediated antitumor cytotoxicity, but also suppresses tumorigenesis through regulating other stromal components in the TME. Further investigations are required to comprehensively delineate the influence of PGRN on stromal cells and the underlying mechanism.

Our findings identify a potential mechanism exploited by tumor cells for immune evasion in PDAC. Future studies on targeting PGRN in combination with other immunomodulatory therapeutic strategies are needed to explore its clinical significance in inducing more durable tumor rejection by targeting central immune escape routes in pancreatic cancer. Given the devastating failure of immunotherapy in PDAC and yet the exciting developments in chimeric antigen receptor T-cell development, PGRN may be a useful target to enhance potent cancer-specific antigen-mediated cytotoxicity.

## Methods

**Clinical specimens**. Expression of PGRN was analyzed in three independent cohorts of patients from the University Hospitals Essen (Essen cohort) and Radboud University Medical Center (Nijmegen cohort) and the phase III adjuvant CONKO-001 randomized trial[44].

For Essen cohort, a retrospective study was carried out according to the recommendations of the local ethics committee of the Medical Faculty of the University of Duisburg-Essen. Clinical data were obtained from archives and electronic health records. In this exploratory retrospective study, a cohort of 54 patients that had undergone pancreatic resection with a final histopathologic diagnosis of human PDAC between March 2006 and February 2016 was used (Approval no: 17-7340-BO).

Additionally, 31 patient samples from an independent cohort from Radboud University Medical Center, Nijmegen, were used to confirm the findings of Essen cohort. The Nijmegen cohort consisted of 31 patients with histologically proven pancreatic ductal adenocarcinoma (PDAC) between November 2004 and January 2015. Importantly, these patients underwent pancreatic resection and the whole tumor was enclosed in large format cassettes allowing to assess of intratumoral heterogeneity. Given the retrospective nature of this study and the anonymized handling of data, informed consent was waived by the medical ethical review board (region Arnhem-Nijmegen) (protocol CMO2018-4420).

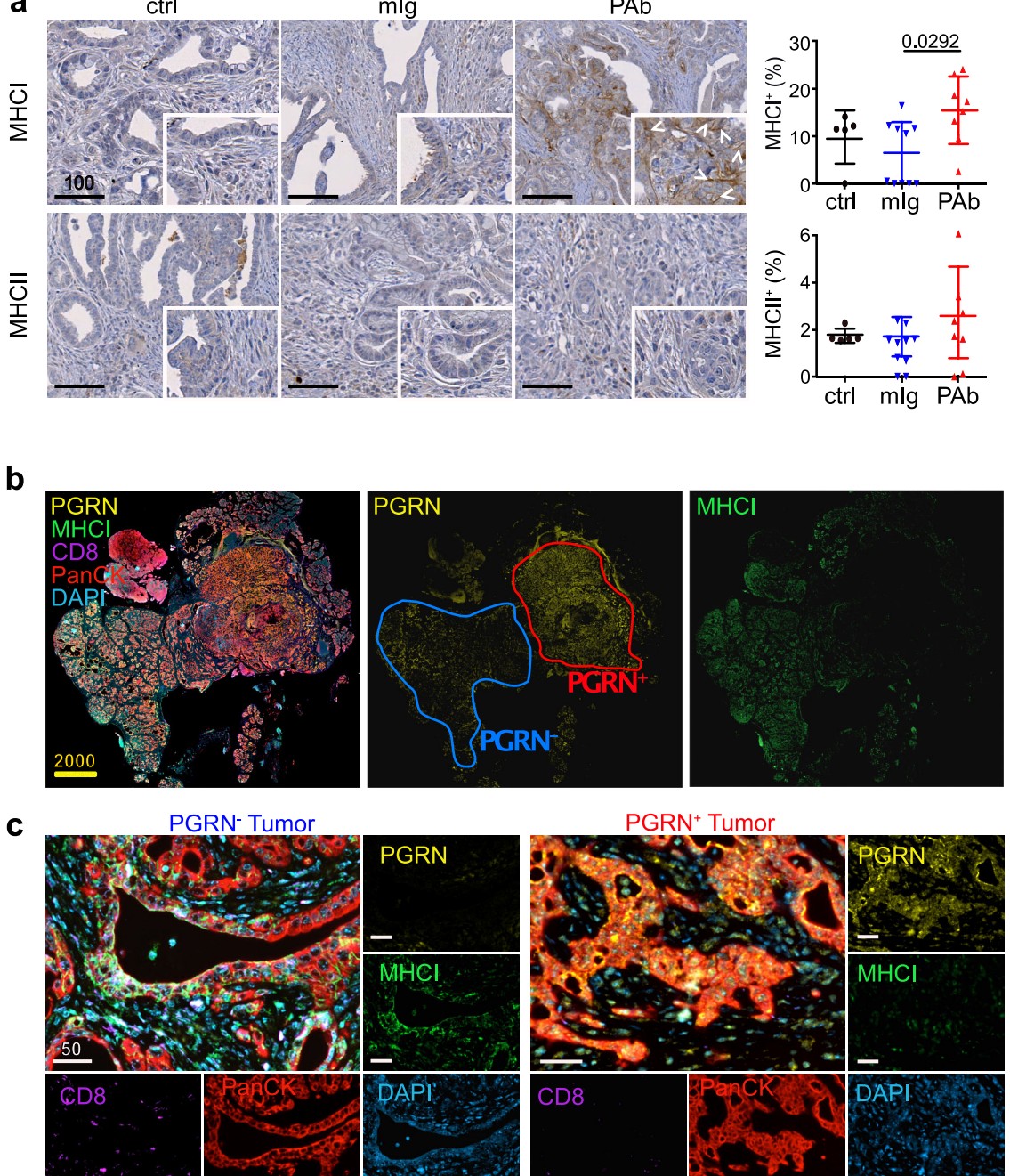

**Fig. 6 In vivo PGRN blockade restores tumor MHCI expression that is spatially associated with increased CD8 cells. a** IHC of MHCI marker H-2Db and MHCII in *CKP* tumors treated with or without PGRN Ab or mIg. ctrl: *n* = 5; mIg: *n* = 10; PGRN Ab (PAb): *n* = 8. The right panels show the percentages of positive cells in the whole tumor. One-way ANOVA, Kruskal–Wallis test. Mean ± SD is shown. **b** mIF of PGRN (yellow), MHCI (green), CD8 (purple) and PanCK (red) in PGRN Ab-treated *CKP* tumors (*n* = 8). Intratumoral heterogeneity of PGRN expression was observed, in which PGRN⁺ and PGRN⁻ regions were depicted in the representative tumor. PGRN and MHCI show opposite staining patterns in the PGRN Ab-treated tumor. **c** mIF showing the differential MHCI and CD8 expression in PGRN⁺ and PGRN⁻ tumor regions. (*n* = 8 PAb-treated *CKP* tumors). Representative images are shown. Scale bar unit: μm.

For CONKO-001, the clinical details of this study have been described previously[44]. In brief, 183 formalin-fixed paraffin-embedded (FFPE) tissue samples of CONKO-001 patients were collected retrospectively. Tissues from 165 patients were suitable for tissue microarray (TMA) construction. To model the existence of intratumoral heterogeneity, three different tumor areas were selected for the construction of TMAs using a manual tissue microarrayer (Beecher Instruments, Wisconsin, USA). Here, we analyzed only the observation arm (*n* = 71), in order to focus on the role of PGRN in PDAC without treatment intervention. The CONKO-001 trial was initiated by the German Study Group for Pancreatic Cancer within the German Cancer Society (Deutsche Krebsgesellschaft), with data collection and trial coordination carried out by the Charité–Universitätsmedizin Berlin, Berlin, Germany. Because adjuvant

chemotherapy with gemcitabine can be administered on an outpatient basis, the participating centers included oncology departments and oncology clinics within hospitals as well as private oncology practices in Germany and Austria. The study was conducted in accordance with the principles of good clinical practice (including regular educational and monitoring procedures), the provisions of the Declaration of Helsinki, and local regulatory requirements. The protocol was approved by the respective institutional review board of each study site. All patients provided written informed consent.

The study was conducted in accordance with the principles of good clinical practice, the ethical principles stated in the current revision of the Declaration of Helsinki, and local legal and regulatory requirements. The protocol was approved by the institutional review board at each study site and all patients provided written informed consent.

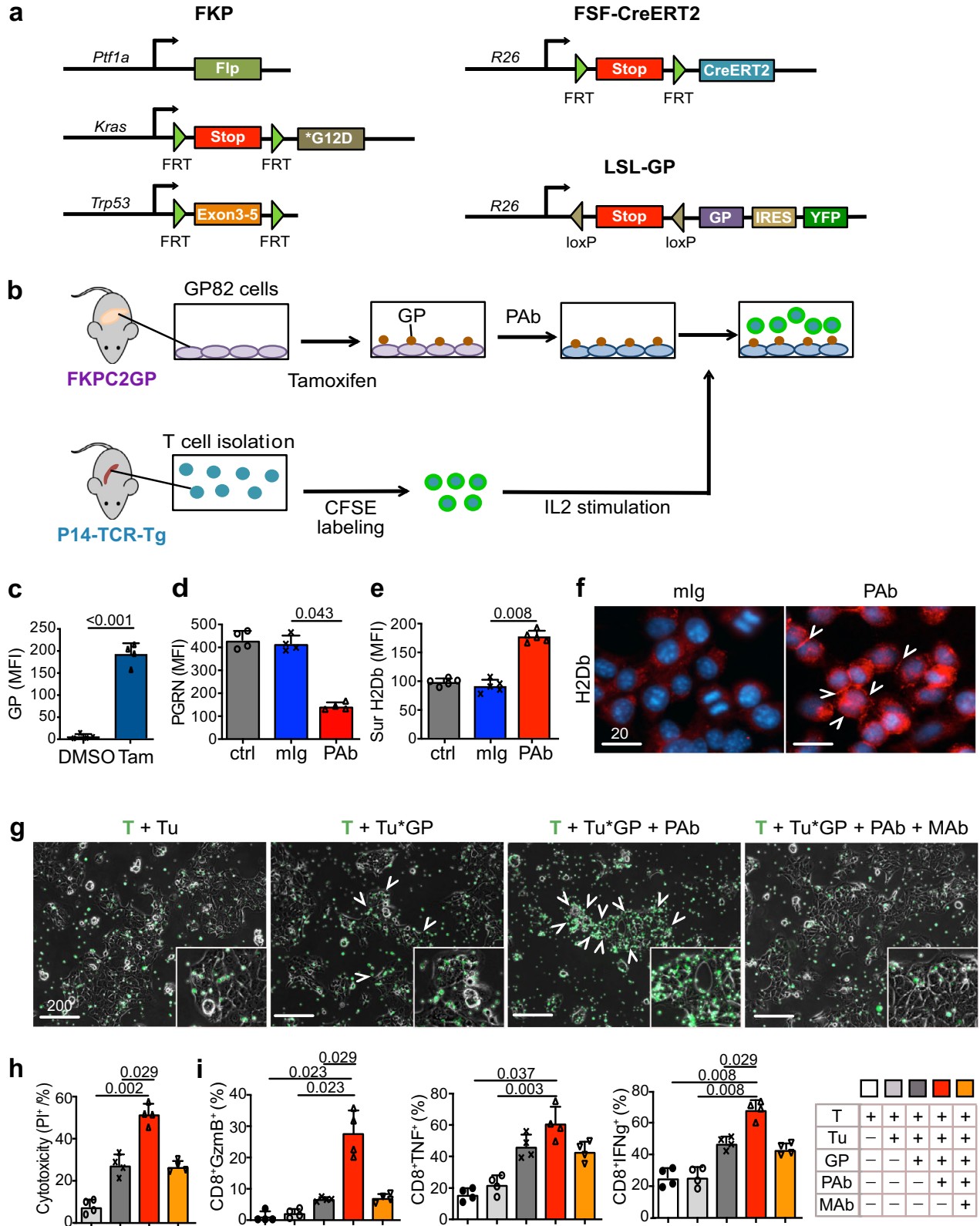

**Gene expression data analysis**. Transcriptional profiling was performed by using a dataset from Maurer et al. (GSE93326), generated from 65 pairs of tumor epithelium and stroma laser capture microdissected samples and 15 bulk tumors[18]. The raw count data of GSE93326 were normalized for library size into counts per million (CPM) by the RLE method implemented in EdgeR (3.28.0) to generate an expression table. We split the data set into stroma and epithelium samples to analyze them separately. Both of them were further divided into *GRN*-high (*n* = 32) and *GRN*-low (*n* = 32) groups by the median CPM value of *GRN*. A differential expression profile was generated by EdgeR comparing the *GRN*-high and *GRN*-low groups (Supplementary Data 3). Gene set enrichment analysis (GSEA) was performed by fgsea (1.12.0) using the default parameters. Enrichment of the Hallmark and the KEGG pathways were tested against the differential

**Fig. 7 In vitro PGRN blockade promotes CD8 antitumor cytotoxicity via MHCI regulation. a** Genetic strategy to induce spatially and temporally controlled GP33 expression by tamoxifen-mediated activation of CreERT2 in pancreatic cells harboring mutant $Kras^{G12D}$ and loss of $Tp53$. $Ptf1a^{wt/flp}$; $Kras^{wt/FSF-G12D}$; $p53^{frt/frt}$ (FKP) mice crossed to $Gt(ROSA)26Sor^{tm3(CAG-Cre/ERT2)Das}$ ($R26^{FSF-CAG-CreERT2}$) and $Gt(ROSA)26Sor^{tmloxP-STOP-loxP-GP-IRES-YFP}$ ($R26^{LSL-GP}$) strains to generate FKPC2GP mice. **b** Experimental setup for co-culture of GP82, LCMV-gp33-expressing cell line derived from FKPC2GP tumor, and the gp33-reactive T cells isolated from the spleen of P14-TCR-Tg mice. **c** LCMV-gp33 (GP) expression in GP82 cells treated with tamoxifen (25 µM) or vehicle control DMSO for 2 days was assessed by flow cytometry ($n = 4$ independent experiments). Two-tailed Mann–Whitney test. Tam: Tamoxifen. **d** Cellular PGRN level in GP82 cells treated with or without PGRN Ab or mIg (100 µg/ml) was assessed by flow cytometry ($n = 4$ independent experiments). One-way ANOVA, Kruskal–Wallis test. **e** Surface expression of MHCI marker H-2Db on GP82 cells treated with or without PGRN Ab or mIg (100 µg/ml) was assessed by flow cytometry ($n = 6$ independent experiments). One-way ANOVA, Kruskal–Wallis test. **f** IF staining of MHCI marker H-2Db of GP82 upon treatment with or without PGRN Ab or mIg (100 µg/ml). White arrowheads indicate the membranous staining of MHCI. ($n = 3$ independent experiments). Representative images are shown. **g** Microscopic images of GP82 cells and LCMV-gp33-reactive T cells (CFSE-labeled, green) after 2 days of co-culture. When anti-MHCI (H-2Db) neutralizing antibody (MHCI Ab, MAb) was included in the treatment, MHCI Ab was added 1 h after PGRN Ab (PAb) treatment. T cells were then added 1 h after MHCI Ab treatment. White arrowheads indicate T-cell clusters accumulated at GP82 cells. ($n = 6$ independent experiments). Representative images are shown. **h** Cytotoxicity level (PI$^+$ %) of GP82 cells upon co-culture with LCMV-gp33-reactive T cells ($n = 6$ independent experiments performed with LCMV-gp33-reactive T cells isolated from six different mice). T: T cells; Tu: GP82 tumor cells; Tu*GP: LCMV-gp33-induced GP82 tumor cells; PAb: PGRN Ab (100 µg/ml); MAb: MHCI Ab (100 µg/ml). One-way ANOVA, Kruskal–Wallis test. For PAb vs Mab: Two-tailed Mann–Whitney test. **i** Percentage of CD8$^+$ cells that are positive for cytotoxic markers granzyme B (GzmB), TNF, and IFNg (% in total T cells) assessed by flow cytometer ($n = 6$ independent experiments performed with LCMV-gp33-reactive T cells isolated from six different mice). One-way ANOVA, Kruskal–Wallis test. For PAb vs Mab: Two-tailed Mann–Whitney test. Mean + SD are shown. *MFI* mean fluorescence intensity. Scale bar unit: µm.

expression profiles ranked by $-\log(p\ value) \times sign(log2FC)$. Pathways with the Benjamini–Hochberg method adjusted $p$ value (padj) smaller than 0.05 were considered significant.

**Cell type estimation using transcriptomes.** Cell type-specific signals were assessed using Syllogist as described by Lu, Dobersalske et al.[19]. The resulting odds ratios were used in inter-sample comparisons to generate hypotheses on the differential content of cell types present in bulk tissues (Supplementary Data 1). To assess the association of each cell type with high and low *GRN* stromal samples, we computed 100× fold random forests for each cell type, similarly to Tan et al.[45] (Supplementary Fig. 1e). The top 20 predictors for *GRN*-high and low samples were selected and are presented in Supplementary Fig. 1f.

**Immunohistochemistry.** FFPE sections were used for all IHC experiments. Antigen retrieval was performed by heat-induced epitope retrieval using citrate buffer (pH6), Tris/EDTA (pH9), or proteinase K treatment. After blocking with serum-free protein blocking solution (Dako), slides were incubated for primary antibodies for 1 hr at RT, a secondary Ab for 30 min at RT, and then subjected to Fast Red or DAB chromogen development. The details of antibodies are shown in Supplementary Table 3. Slides were then counterstained with hematoxylin, dehydrated, and mounted. Stromal content and acinar cells were assessed by Movat's pentachrome staining following the manufacturer's protocol (modified according to Verhoeff, Morphisto GmbH, Germany).

Slides were scanned and digitalized by Zeiss AxioScanner Z.1 (Carl Zeiss AG, Germany) with ×10 objective magnification. The percentage of positive cells for IHC staining was quantified by Definiens (Definiens AG, Germany). For quantification, the total number of cells of the whole-tissue section was measured based on nuclei staining (hematoxylin) detected by the software (Supplementary Fig. 11).

**Multiplexed IF histological staining.** Multiplexed IF was performed using the Opal multiplex system (Perkin Elmer, MA) according to the manufacturer's instruction. In brief, FFPE sections were deparaffinized and then fixed with 4% paraformaldehyde prior to antigen retrieval by heat-induced epitope retrieval using citrate buffer (pH6) or Tris/EDTA (pH9). Each section was put through several sequential rounds of staining; each includes endogenous peroxidase blocking and non-specific protein blocking, followed by primary Ab and corresponding secondary horseradish peroxidase-conjugated polymer (Zytomed Systems, Germany or Perkin Elmer). Each horseradish peroxidase-conjugated polymer mediated the covalent binding of different fluorophores using tyramide signal amplification. Such covalent reaction was followed by additional antigen retrieval in heated citric buffer (pH6) or Tris/EDTA (pH9) for 10 min to remove antibodies before the next round of staining. After all sequential staining reactions, sections were counterstained with DAPI (Vector lab). The sequential multiplexed staining protocol is shown in Supplementary Table 4. Slides were scanned and digitalized by Zeiss AxioScanner Z.1 (Carl Zeiss AG, Germany) with 10x objective magnification. Quantification of individual and/or co-expressing markers in the mIF images was performed with HALO (Indica Labs, Albuquerque, NM, USA) (Supplementary Fig. 12).

**Spatial imaging analysis.** An intensity threshold was used to generate masks for each fluorescent channel and the binary information for cellular and nuclear signals was coregistered. Automated analysis of cell distance was performed by a Java and R-based algorithm. Using ImageJ, overlapping mask regions were employed to identify cells, which were marked with a point at the center of the DAPI$^+$ cell nucleus.

The R Package Spatstat[1] was used to convert cell coordinates to Euclidian distances between cells of interest. The algorithm assesses the average distance and number of unique neighbors between any two cells. The threshold to pair up two cells was set to 50um. The algorithm gives the percentage of MHCI$^+$ cells in PGRN$^+$/PanCK$^+$ and PGRN$^-$/PanCK$^+$ populations, as well as the number of pairs between CD8$^+$ and PGRN$^+$/MHCI$^-$ or PGRN$^-$/MHCI$^+$ cells in each image[46].

The above spatial interaction analysis was validated with HALO (Indica Labs) at the whole-tissue scale (Supplementary Fig. 12f).

**Mouse strains and tumor models.** Animal experiments were approved under license number 84-02.04.2017.A315 by the Landesamt für Natur, Umwelt und Verbraucherschutz Nordrhein-Westfalen. All animal care and protocols adhered to national (Tierschutzgesetz) and European (Directive 2010/63/EU) laws and regulations as well as European Federation of Animal Science Associations (FELASA) http://www.felasa.eu/. $Ptf1a^{wt/Cre}$; $Kras^{wt/LSL-G12D}$; $p53^{fl/fl}$ (CKP) mice have been described previously[35]. All mice are on C57BL/6 background. FKPC2GP is generated by crossing $Ptf1a^{wt/Flp}$; $Kras^{wt/FSF-G12D}$; $p53^{frt/frt}$ (FKP) mice to $Gt(ROSA)26Sor^{tm3(CAG-Cre/ERT2)Das}$ ($R26^{FSF-CAG-CreERT2}$) and $Gt(ROSA)26Sor^{tmloxP-STOP-loxP-GP-IRES-YFP}$ ($R26^{LSL-GP}$)[47] strains.

All animals were numbered, genotypes were identified and animals were then assigned to groups for analysis. For treatment experiments mice were randomized. None of the mice with the appropriate genotype were excluded from this study. Details of original and interbred mouse strains are listed in Supplementary Table 5a, b[48–54].

**Cell culture and treatments.** Human PDAC cell lines, PaTu8988T, MiaPaCa2, and HupT4, were purchased from the American Type Culture Collection. Stable cell lines for GRN suppression were established by transfecting *GRN* shRNA into Patu8988T and MiaPaCa2. Scramble shRNA was included as negative control (nc) for transfection. Sequences of shRNA and nc are as follows,

sh_GRN1:
AGGCCCTGATAGTCAGTTCGAATtgacaggaagATTCGAACTGACTATCAGGGC

sh_GRN2:
AGGAAGGACACTTCTGCCATGATtgacaggaagATCATGGCAGAAGTGTCCTTC

sh_GRN3:
AGGTGACCTGATCCAGAGTAAGTtgacaggaagACTTACTCTGGATCAGGTCAC

nc:
AGGGAATCTCATTCGATGCATACtgacaggaagGTATGCATCGAATGAGATTCC

All transfectants were maintained in 10% FBS-supplemented Dulbecco's Modified Eagle Medium (DMEM) with 2 mg/mL of G418 (Life Technologies, Thermo Fisher Scientific, MA). Exogenous PGRN stimulation was performed by incubating HupT4 with or without 0.4 ng/mL rPGRN for 1 day.

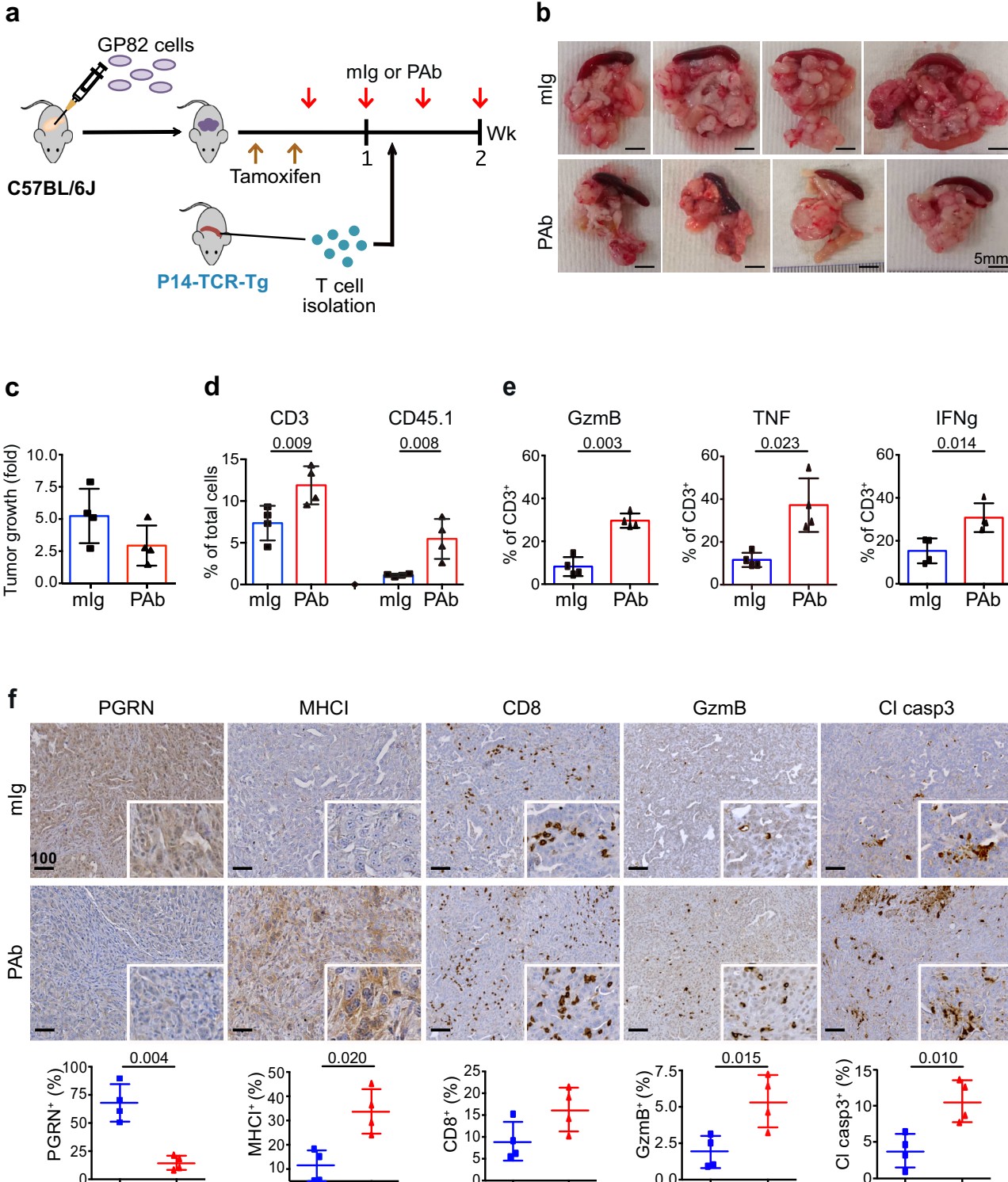

**Fig. 8 In vivo PGRN blockade promotes antigen-specific T-cell cytotoxicity against tumors in orthotopic FKPC2GP model. a** Timeline for treatment of anti-PGRN antibody (PAb) or mIg (50 mg/kg) in an orthotopic model of GP82 cells in C57BL/6 J mice. **b** Tumors and spleens of mIg-treated ($n = 4$) and PGRN Ab-treated ($n = 4$) mice with orthotopic GP82 transplantation and intravenous injection of LCMV-gp33-reactive T cells freshly isolated from P14-TCR-Tg mice. **c** Tumor growth was assessed by ultrasound imaging and presented as a fold change in tumor volume before and after PGRN Ab (PAb) or mIg treatment started. **d** Tumors were digested into disaggregated cells and stained for T-cell infiltration. Flow cytometric analysis showing the percentage of tumor-infiltrating T (CD3+) cells and the exogenously injected LCMV-gp33-reactive (CD45.1+) T cells, in tumors treated with PGRN Ab ($n = 4$) or mIg ($n = 4$). Two-tailed Mann–Whitney test. **e** Flow cytometric analysis showing the expression of cytotoxic markers granzyme B (GzmB), TNF, and IFNg on CD3+ T cells in tumors with PGRN Ab ($n = 4$) or mIg ($n = 4$). Two-tailed Mann–Whitney test. **f** IHC staining of PGRN, MHCI marker H-2Db, CD8, GzmB, and cleaved caspase 3 (Cl Casp3) in orthotopic GP82 tumors with PGRN Ab ($n = 4$) or mIg ($n = 4$). The lower panels how the percentages of positive cells in the whole tumors quantified by Definiens. Two-tailed Mann–Whitney test. Mean ± SD is shown. Scale bar unit: μm.

For GP82, it was established from the tumor of an FKPC2GP mouse no. 82. After enzymatic digestion of the tumor, the desegregated cells were grown and maintained in 10% FBS-supplemented DMEM. After 3rd passage, the cell expression of epithelial marker EpCAM and fibroblast marker α-SMA was assessed by immunofluorescence staining and flow cytometry, and confirmed enrichment of epithelial cells with <1% contamination of fibroblasts.

**IF staining and flow cytometric analysis**. For intracellular PGRN expression, cells were permeabilized with ice-cold 0.1% saponin and then incubated with mouse anti-human PGRN ([10,55]), or an equal amount of corresponding isotype control, followed by FITC-goat anti-mouse Ab (BD biosciences). CD3, CD8, CD45.1 on T cells, cells were stained with corresponding antibodies or equal amount of corresponding isotype controls. For intracellular granzyme B, TNF, and IFNg in T cells, or LCMV-gp33 in tumor cells, cells were fixed with 4% paraformaldehyde for 10 min at 37 °C. After washing twice with PBS, cells were permeabilized with 0.1% Saponin for 20 min and then stained with antibodies and corresponding isotype. Details of primary antibodies are listed in Supplementary Table 6. Cells were then washed, resuspended, and subjected to analysis. Expression of corresponding molecules of 10,000 viable cells was analyzed by flow cytometry (FACSCelasta; BD Biosciences) as mean fluorescence intensity. Raw data were analyzed using FlowJo software version 7.5.5 (Tree Star Inc., Ashland, OR).

**Enzyme-linked immunosorbent assay to measure soluble PGRN level**. Soluble PGRN levels in human plasma samples and culture supernatants were detected by a human PGRN ELISA kit (Adipogen Inc.). Plasma samples from normal individuals and PDAC patients were diluted at 1:100 using the diluent provided by the kit, while culture supernatants from in vitro experiments were undiluted. While for mouse plasma collected from in vivo Ab treatment experiment, plasma PGRN levels were measured by mouse PGRN ELISA kit (Adipogen). Plasma samples were diluted at 1:5 using the diluent provided by the kit.

**In vivo Ab treatment on CKP**. CKP mice were injected with or without mouse immunoglobulin (mIg) or anti-PGRN Ab (clone A23[32]) twice weekly at 50 mg/kg in PBS, via intraperitoneal injection at four weeks of age for two consecutive weeks. Mice were weighed twice weekly throughout the experiment. Mice with >15% body weight loss were terminated even the endpoint was not reached. At the endpoint, serum, tumors, and organs were collected and processed for histological and immunohistochemical analysis. Tumor burden was measured by establishing the gross wet weight of the pancreas/tumor.

In the CD8[+] T-cell depletion experiments, 25 mg/ml of anti-CD8 depleting Ab (clone 2.43, Bio X Cell) was injected i.p at days −3 and day 1 relative to the starting date of PGRN Ab treatment.

**Real-time quantitative reverse-transcription polymerase chain reaction**. Real-time quantitative PCR (qPCR) was performed by Roche LightCycler® 480 using LightCycler® 480 SYBR Green I Master Kit (Roche GmbH, Germany). Primers for PGRN (GRN) were designed using the NCBI Primer Blast and purchased from Eurofins MWG Operon GmbH, Ebersberg, Germany. Primers and probes for PGRN were GRN-forward (5′-CAAATGGCCCACAACACTGA-3′), GRN-reverse(5′-CCCTGAGACGGTAAAGATGCA-3′), and GRN-probe (5′-6FAMC-CACTGCTCTGCCGGCCACTCMGBNFQ-3′). Real-time qPCR experiments were run under 58 °C annealing conditions and amplification was run for 45 cycles. A melting curve was implemented in each experiment to prove single product amplification. Data were analyzed using ΔCt calculations where GAPDH or GUSB served as housekeeper control for normalization. The amplification efficiency was experimentally assessed or assumed as 2 (doubling each cycle). Relative mRNA expression levels compared to housekeeper gene expression (efficiency-ΔCt) were used for visualization.

**Immunofluorescence staining**. Cells were grown on eight-well cell culture chamber slides (Lab-Tek, Waltham, USA) for 3 days, slides were then washed with PBS three times and then fixed and permeabilized with ice-cold methanol for 5 min. After washing with PBS three times, slides were blocked with 10% normal goat serum in PBS for 1 h. Cells were incubated with a 1:100 dilution of the corresponding primary antibodies (Supplementary Table 7) in 1% BSA at RT for 1 hr. Slides were washed and incubated with the secondary Ab (1:400, Thermo Scientific, Waltham, USA) for 1 h at room temperature in the dark. Afterwards, slides were washed three times with PBS, counterstained with DAPI (Vector Laboratories, Burlingame), and scanned with AxioScanner.

**Western blot analysis**. Protein lysates from tumor cell lines were prepared in RIPA Buffer (Cell Signaling Technologies, Danvers) with protease and phosphatase inhibitor cocktail (Cell Signaling technologies). After quantification of protein concentrations by BCA protein assay (Thermo Scientific), protein samples were resolved on 8–10% SDS polyacrylamide gels and then transferred to nitrocellulose membranes (Bio-Rad, Hercules) with Trans-Blot Turbo Transfer System (Bio-Rad). Membranes were blocked with 5% BSA for 1 h at room temperature and then incubated with primary antibodies (LC3B: Sigma, 1:1000; p62: cell signaling,

1:1000; Actin: Sigma, 1:10,000) at 4 °C overnight. Anti-rabbit secondary Ab was diluted at 1:25000 in blocking buffer and incubated for 1 h at room temperature. Signals were developed by SuperSignal Chemiluminescence (Thermo Scientific) and visualized with ChemiDoc MP Imaging System (Bio-Rad).

**Lysosomal staining**. Lysosomal staining was performed using Cytopainter Lyso-Green indicator reagent (Abcam) according to the manufacturer's instructions. Briefly, cells were grown to ~70% confluence and then incubated with Cytopainter LysoGreen indicator reagent (Abcam) for 1 h at 37 °C with 5% CO2. Cells were then washed twice in pre-warmed Hank's balanced salt solution (HBSS, Life Technologies), trypsinized with 0.05% trypsin EDTA (Technologies), and resuspended in 1× phosphate-buffered saline (Life Technologies) for flow cytometry. Lysosomal levels were measured by a flow cytometer (BD).

**LysoSensor DND-189 staining**. Cells were loaded with 1 μM LysoSensor Green DND-189, which is more fluorescent in acidic environments and less fluorescent in alkaline environments, and incubated for 30 min at 37 °C. The cells were fixed with 4% PFA for 20 min and washed, and then fluorescence images were captured using AxioScanner (Zeiss). The average area of DND-189[+] vesicles of 30 cells of each treatment was measured by ZEISS ZEN 2.3 software.

**DQ-BSA assays for proteolytic activity in lysosomes**. Cells were seeded onto 24-well plates and then treated with or without Balfinomycin A (100 nM, Sigma) for 24 h. The cells were pulse-labeled with DQ-BSA conjugated with Alexa Fluor 488 (20 μg/mL, Thermo Fisher Scientific) for 30 min. The DQ-BSA pulse labeling was followed by PBS and the corresponding chase period (0, 1, 2, 4 h). Cells were then washed again with PBS, trypsinized, and fixed with 4% PFA for 10 min. Cells were kept in cold PBS prior to FACS analysis (BD FACSCalibur). The FACS data were analyzed using FlowJo software.

**Isolation of T cells from P14-TCR-Tg mice**. Spleens harvested from P14-TCR-Tg mice were homogenized, and then lysed by Ammonium-Chloride-Potassium lysis buffer. T cells were negatively selected by MACS (Miltenyi Biotec) according to the manufacturer's instructions. After isolation, T cells were labeled with 5 uM CFSE (Thermofisher), and resuspended 10% FBS-supplemented RPMI medium in pre-stimulated with 20 ng/ml IL-2 (Peprotech) for 1 hr, before co-culture with GP82 cell line or orthotopic injection in C57BL/6 J mice.

**Co-culture of LCMV-gp33-reactive T cells and GP82 cells**. GP82 cells were treated with or without tamoxifen to induce LCMV-gp33 expression for at least 5 days, and then treated with or without PGRN Ab or mIg to suppress PGRN levels. After 2-day treatment, cells were harvested and seeded to six-well with 10% FBS-supplemented DMEM and were ready to co-culture. Then, LCMV-gp33-reactive T cells were isolated, and then cultured with or without GP82 cell lines at ratio of 8:1, in 10% FBS-supplemented DMEM for 2 days. Cells were then photographed under a microscope and then harvested for antitumor cytotoxicity by propidium iodide (PI) staining. Besides, T-cell activity was assessed by staining for CD8, Granzyme B, TNF, IFNg by flow cytometry.

**In vivo Ab treatment on orthotopic model of GP82 cells in C57BL/6 J mice**. GP82, primary cell line derived from one of the FKPC2GP tumor, was transplanted orthotopically into the pancreas of C57BL/6 J mice with needle injection under ultrasound imaging guidance. After orthotopic transplantation of GP82 cells, ultrasound imaging was performed once a week to monitor tumor growth. Once tumor volume reached 100 mm³, intraperitoneal injection of tamoxifen (75 mg/kg) will be performed to induce LCMV-gp33 expression. After the second injection of tamoxifen (2 days after the first injection), mouse immunoglobulin (mIg) or anti-PGRN Ab (clone A23[32]) twice weekly at 50 mg/kg in PBS, via intraperitoneal injection for 2 weeks. After the first treatment of A23 or mIg, LCMV-gp33-reactive T cells isolated from P14-TCR-Tg mice were injected intravenously. Mice were weighed twice weekly throughout the experiment, and tumor growth was monitored by ultrasound imaging once a week. Mice with >15% body weight loss were terminated even the endpoint was not reached. At the endpoint, serum, tumors, and organs were collected and processed for immunohistochemical and flow cytometric analysis.

**Statistical data analysis**. All statistics were performed using GraphPad Prism 6.0 (GraphPad Software, La Jolla, CA) and SPSS v.22 (IBM SPSS Statistics, Armonk, NY). Survival data were analyzed by Log-rank (Mantel–Cox) test, while correlation analysis was by Spearman's rank correlation coefficient. The Chi-square test was used to assess the independence between two categorical variables. Two-tailed Mann–Whitney test was applied for non-normally distributed data comparison between two groups. For multiple group comparison, the Kruskal–Wallis test was used. Data are represented as mean ± S.D. The exact $p$ values are shown in the figures.

**Reporting summary**. Further information on research design is available in the Nature Research Reporting Summary linked to this article.

## Data availability

The data used to analyze for transcriptomic profiling of *GRN*-high and low tumor epithelium and stroma is available in Gene Expression Omnibus (GEO) database under accession code GSE93326. All other data are available in the article and its Supplementary files or from the corresponding author upon reasonable request. Source data are provided with this paper.

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

## Acknowledgements

P.F. Cheung has received funding from the Deutsche Forschungsgemeinschaft (DFG) for this study (CH 2320/2-3). J.T. Siveke is supported by the German Cancer Consortium (DKTK), the Deutsche Forschungsgemeinschaft (DFG) through grant SI1549/3-1 (Clinical Research Unit KFO337), SI1549/4-1 and the Collaborative Research Center SFB824 (project C4), the Deutsche Krebshilfe (German Cancer Aid) through #70112505, PIPAC and #70113834, PREDICT-PACA, and the Wilhelm-Sander Stiftung (grant number: 2019.008.1, joined with M. Trajkovic-Arsic). S.T. Cheung is supported by the Health and Medical Research Fund (05160556). E.H.J.G. Aarntzen received funding from Radboud Oncology Fund/Bergh in het Zadel (KUN2016-8106) and Radboud Oncology Fund/Barghse Jongens (ROF1817). C.W. Yip is supported by the Ministry of Education, Culture, Sports, Science and Technology (MEXT) funding to RIKEN Integrative Medical Sciences (IMS). I. Cima and B. Scheffler received funding from the Wilhelm-Sander Stiftung (grant number: 2017.148.1). B. Scheffler is also supported by the German Cancer Consortium (DKTK), the Deutsche Forschungsgemeinschaft (DFG) through grant SCHE656/2-1 (Clinical Research Unit KFO337). A. Paschen is supported by the Deutsche Forschungsgemeinschaft (DFG, German Research Foundation; PA 2376/1-1 [KFO 337]). We thank Dr. T. Ebel (Zentrum f. Pathologie Essen-Mitte) for kind support. We would like to thank Dr. T. Jacks, Dr. J. Jonkers, Dr. A. Berns, Dr. D. Tuveson, Dr. D. Kirsch, Dr. D. Saur, Dr. R.M. Schmid, Dr. D.G. Kirsch and Dr. D. Merkler for providing transgenic animals. Support of the DKFZ Genomics and Proteomics Core Facility is gratefully acknowledged. We are thankful to the West German Biobank Essen for infrastructural support.

## Author contributions

Concept and design: P.F.C., S.T.C., J.T.S. Acquisition of data: P.F.C., J.Y., R.F., E.H.L.S., A.B., K.S., E.M.M.S. performed histological stainings; P.F.C., A.B., E.H.L.S., K.K., X.Z. performed in vitro PGRN modulation and functional assays; P.F.C., J.Y., R.F., A.B., A.S., K.S., K.A., C.N., A.B. performed in vivo and in vitro experiments. C.W.Y., L.W.C.N., S.T.C. contributed to PGRN Ab, recombinant protein, and plasmids, and assay establishment. Analysis and interpretation of data: C.W.Y., S.L., S.S.L., I.C., B.S. performed bioinformatic analysis and assisted data interpretation. C.S., D.R.E. performed computational spatial analysis and assisted data interpretation. P.F.C., E.H.L.S., J.J.Y. performed quantitative analysis for IHC and mIF. D.R.E., C.S. performed computational spatial analysis on mIF stainings. K.L. and L.C. assisted the data interpretation of LCMV-gp33-expressing genetic PDAC model systems. J.K.S., M.S., E.H.J.G.A., M.B., U.P., H.O., P.M., A.W.C., J.T.S. assisted evaluation of clinical-pathological data of patients. P.F.C., J.T.S., H.C., A.P. interpreted the results of experiments. Writing and revision of the manuscript: P.F.C. and J.T.S. wrote the manuscript. All authors commented on the manuscript and approved the final version. Study supervision: P.F.C., S.T.C., J.T.S.

## Funding

## Competing interests

J.T.S. reports the following disclosures: Bristol Myers Squibb, Celgene, Roche (Research Funding); AstraZeneca, Bayer, Bristol Myers Squibb, Celgene, Immunocore, Novartis, Roche, Shire (Consulting or advisory role); AstraZeneca, Aurikamed, Baxalta, Bristol Myers Squibb, Celgene, Falk Foundation, iomedico, Immunocore, Novartis, Roche, Shire (honoraria); minor equity in iTheranostics and Pharma15 (<3%); member of the Board of Directors for Pharma15, all outside the submitted work. The other authors declare no competing interests.
