## [Peer Review File · Nature Communications]

Progranulin mediates immune evasion of pancreatic ductal adenocarcinoma through regulation of MHCI expressionEditorial Note: Parts of this Peer Review File have been redacted as indicated to maintain the confidentiality of unpublished data.

REVIEWER COMMENTS

Reviewer #1 (Remarks to the Author):

The manuscript submitted by Cheung and collaborators is a very nice and well conducted study showing that cancer cell progranulin (PGRN) drives immune evasion in PDAC. Through a wide experimental approach, technically sound and displaying robust data the authors demonstrate the role of PGRN in PDAC immunogenicity and show that PGRN blockade restores tumor immunogenicity and antigen-specific cytotoxicity in a spontaneous mouse model expressing LCMV-gp33 antigen.

Comments:

There is a recent paper showing that KRAS drives immune evasion in a mouse model and identify BRAF and MYC as key mediators (Ischenko Nat. Commun. 2021 12:1482). Activating KRAS mutations are found in 90% of PDAC patients. However, It will be interesting to know whether PGRN-driving immune evasion was independent of KRAS status and/or whether there are any evidences that could sustain KRAS and PGRN acting through cooperative mechanisms for immune escape in PDAC.

In the in vivo experiment of PGRN blockade, authors examined the abundance of fibroblasts in the tumor stroma. They conclude that there were no variations in the tumor fibrosis based on the staining of alpha-SMA. In this study the authors predominantly analyzed the myofibroblast subtype of cancer associated fibroblasts (myCAF). However, in line with the heterogeneity of CAF subtypes, authors should analyze whether PGRN blockade was triggering any effect in the inflammatory iCAF population.

Minor points:

Figure 1a. PGRN staining is difficult to appreciate in the representative images, particularly in High grade PanIN and PDAC.

Was there any specific reason to detect MHC I in mice experiments recognizing only the H-2Db class I molecule?. Did the authors analyze effects on H-2Kb class I?

Materials and Methods: There is no need to specify the method of euthanasia used in the study.

Reviewer #2 (Remarks to the Author):

In this manuscript the authors unveiled a novel function of progranulin (PGRN) in the regulation of MHC I in PDAC. They demonstrate that tumor cell-derived PGRN drives MHC I downregulation in both murine tumors and human biopsies, correlating with lower CD8 infiltration. In addition, they demonstrate that PGRN blockade in vivo increases CD8 cytotoxicity and tumor infiltration. This effect is lost with simultaneous blockade of MHC I. The manuscript is well written, science is novel and methods and statistics are appropriate. This reviewer has the following comments:

1. While PGRN blockade had an effect on tumor progression in a genetic PDAC model, it did not in tumors expressing LCMV-gp33 in the absence of adoptive transfer of gp33-TCR Tg T cells. What are the possible reasons for such discrepancy?
2. Does PGRN blockade have any effect on long-term survival in these models?
3. To really establish CD8 TILs as the main effectors upon PGRN blockade, authors should provide survival and/or antitumor data with CD8 depletion in a PDAC model.

Reviewer #3 (Remarks to the Author):

The study by Cheung and colleagues describe a role for tumor cell expressed progranulin in

regulation of MHC-I and immune evasion in pancreatic ductal adenocarcinoma. The authors show that increased PGRN expression in tumor epithelial cells increases with stage of malignancy and correlates with poor patient prognosis. Interestingly, macrophage-associated PGRN does not correlate with patient outcomes. Further, PGRN expression levels were shown to correlate with MHC-I expression and T cell infiltration in PDAC tumors – where PGRN low cells show higher MHC-I expression and increased T cell association. Accordingly, knockdown of PGRN or treatment with an PGRN blocking antibody is able to increase MHC-I levels and induce greater activated CD8+ T cell infiltration in FKP mouse model of PDAC. Mechanistically, the authors provide some data to suggest that PGRN suppression inhibits autophagy – which was recently shown to regulate MHC-I levels and surface expression in PDAC.

Finally the authors develop and utilize a new model antigen mouse model to further test the efficacy of utilizing an anti-PGRN ab for enhancing MHC-I expression, increasing T cell infiltration and blocking tumor growth. While a significant change in tumor growth was not observed in this model, increase MHC-I and T cell association as seen in the FKP studies were recapitulated. Overall this is a rigorous and well executed study that highlights a previously unrecognized role for tumor cell autonomous PGRN in modulation of immune evasion in PDAC. However, the lack of clear mechanism connecting PGRN and MHC-I is a weakness of the study. Moreover, the claim that tumor cell autonomous effects exclusively effect MHC-I and immune evasion is not entirely justified given that ECM, stromal cell, endothelial cell derived PGRN are not considered. Therefore, my suggestions are mostly focused on strengthening these aspects of the manuscript.

1. If the authors wish to establish PGRN as an upstream regulator of autophagy in PDAC they should complement the data shown in figure 3 with autophagy flux assays which are standard in the field. These include use of the dual fluorescent LC3 autophagy reporter or western blot analysis of LC3 lipidation in the presence/absence of lysosome blockade in the PRGN WT/KD setting.
2. Likewise, if the authors believe that PRGN loss impacts lysosome function then appropriate assays to test lysosome activity should be included (pH measurements, ability to degrade cargo, diameter measurements etc). For example, GRN^{-/-} microglia show lysosomal defects (PMC4860138). It would be interesting to know whether PGRN suppression in PDAC display similar defects.
3. It is unclear why the authors chose to conduct the shPGRN KD experiments in combination with serum starvation (Fig. 3c). Can the authors repeat these experiments in normal media conditions to definitely establish serum independent PGRN effects on autophagy?
4. It is unclear the extent to which PGRN is secreted from PDAC cells versus intracellular localized and how this compares to macrophages or other cellular populations within the tumor microenvironment. Accordingly, is PGRN secretion important for its role in modulating MHC-I levels? Or is the function of PRGN in regulation of MHC-I restricted to an intracellular role?
5. The authors show in Fig. S2a that human PDAC cells show differential PGRN expression levels. How does this correlate with MHC-I levels and localization given that MHC-I is low across PDAC?
6. It would be beneficial to provide quantification of the level of PRGN at different stages of PDAC progression (Fig.4a) in order to justify the statement that “as lesions progressed to early PDAC, PGRN levels reached the maximum and slightly decreased in advanced stage” – line 197.
7. What is the overall survival of FKP mice with or without PGRN antibody treatment? This information would be very useful to include.
8. In figure 5 it is difficult to assess whether the increase levels of activation marks are simply reflective of increased infiltration of CD8+ T cells. It would be useful to establish % co-staining between CD8 and the activation markers to gain a clearer indication of the percentage of total CD8 cells that are also positive for granzyme B, Eomes, T-bet. This would justify the statement “PGRN Ab enhanced the cytotoxicity of the infiltrating CD8+ T cells”
9. Can the authors confirm via flow cytometry that anti-PGRN (KD or antibody treatment) leads to

increased cell surface MHC-I in vitro or better yet in vivo?

10. It is confusing as to why gp33 expressing tumors treated with PGRN Ab but without LCMV-gp33 reactive T cell injection do not show significant native CD8 T cell infiltration (line 319-322). The authors show that the tumor cells elevate MHC-I (Fig S7d), which should be sufficient to elicit increased native T cell infiltration against endogenous antigen as in the FKP model, and lead to some level of anti-tumor response. Could the authors explain why this does not occur? Overall the data generated utilizing the model antigen system, while useful, is less compelling than the results obtained using the FKP model. This is also reflected in the absence of a significant decrease in overall tumor volume T/Tu/GP/Pab arm.

Minor comments:

Clear definition of abbreviations for Figure 7g and H should be indicated in the legend. Statistical testing should be calculated between the T/Tu/GP vs T/Tu/GP/P-AB treatment arms in order to establish whether blocking PGRN can elevate T cell activation and anti-tumor cytotoxicity.

In Fig. 2c MHC-I low status in PRGN negative section correlates to areas in which PanCK staining is also low. Is this representative and if so can the authors explain this?

Line 331 of the discussion relating to "antigenic tumor peptides or dendritic cells loaded with shared peptides" requires references.

Line 334 of the discussion which states that PRGN in tumor cells is "a key instructive regulator of immune evasion" is overstated given that the exact mechanism as to how this occurs is not established and the data showing a block in autophagy is not convincing. This sentence should be revised if additional data in support of a role for PGRN in regulation of autophagy in PDAC is not provided.

RESPONSES TO REVIEWER COMMENTS

Reviewer #1 (Remarks to the Author):

The manuscript submitted by Cheung and collaborators is a very nice and well conducted study showing that cancer cell progranulin (PGRN) drives immune evasion in PDAC. Through a wide experimental approach, technically sound and displaying robust data the authors demonstrate the role of PGRN in PDAC immunogenicity and show that PGRN blockade restores tumor immunogenicity and antigen-specific cytotoxicity in a spontaneous mouse model expressing LCMV-gp33 antigen.

Comments:

→ There is a recent paper showing that KRAS drives immune evasion in a mouse model and identify BRAF and MYC as key mediators (Ischenko *Nat. Commun.* 2021 12:1482). Activating KRAS mutations are found in 90% of PDAC patients. However, It will be interesting to know whether PGRN-driving immune evasion was independent of KRAS status and/or whether there are any evidences that could sustain KRAS and PGRN acting through cooperative mechanisms for immune escape in PDAC.

Response: We appreciate the reviewer for the very insightful comments and suggestion. The immunological impact caused by KRAS deficiency in the transplanted mouse model of PDAC by Ischenko *et al* are indeed quite similar to PGRN blockade in our spontaneous PDAC mouse model, particularly the augmented T cell infiltration and MHC upregulation.

We have investigated the association of PGRN expression with KRAS status in the CONKO-001 cohort. In the control arm of CONKO-001, KRAS status of 46 patients was identified (10 wild type and 36 mutated). Correlation analysis was performed on the KRAS mutation status with PGRN expression in tumor cells, immune cells, or both. However, no significant association can be observed (**Table S5**). Next, we assessed the expression of MYC, the key mediator for KRAS-driven immune evasion as reported by Ischenko *et al*, in human PDAC cell lines modulated for PGRN expression. We observed no change in myc expression upon both PGRN suppression or stimulation (**Figure S10a**). Notably, in the same study, TGF- β signaling/Smad4 signaling was found to play different roles from MYC in pancreatic tumor maintenance in their model. The author suggested that the role of TGF- β signaling might serve as an escape mechanism from oncogenic KRAS addition in PDAC development. Interestingly, we found TGF- β 1 secretion to be significantly reduced in *GRN*-suppressed PDAC cells (**Figure S10b**). Besides, a TGF- β gene signature was significantly enriched in *GRN*-high tumors of the Maurer *et al* dataset¹ (**Figure 1**), implying a potential link between PGRN-induced immunomodulation and TGF- β signaling. However, given the complexity of the TGF- β pathway, further in-depth research is required to determine the interaction among PGRN, TGF- β and KRAS signaling in regulating immune evasion in PDAC.

We have discussed the potential interaction among PGRN, KRAS and TGF- β in the "Discussion" section (page 9, line 400-418).

Besides, in addition to T cell infiltration and MHC expression, KRAS drives a wide range of immune evasion activities including macrophages, CAFs, etc. While PGRN might share certain common mechanisms with KRAS-driven pathway to elicit its immunomodulatory effects, we would like to focus in this manuscript specifically on the role of PGRN in anti-tumor cytotoxicity of CD8 cells and tumor MHC1 modulation,

which we have provided solid evidence in the revised version for the underlying mechanism focusing on lysosomal and autophagy pathway (**Figure 3, S3, S4**).

The corresponding new data is described in “Results” section (page 5-6, line 166-216).

→ *In the in vivo experiment of PGRN blockade, authors examined the abundance of fibroblasts in the tumor stroma. They conclude that there were no variations in the tumor fibrosis based on the staining of alpha-SMA. In this study the authors predominantly analyzed the myofibroblast subtype of cancer associated fibroblasts (myCAF). However, in line with the heterogeneity of CAF subtypes, authors should analyze whether PGRN blockade was triggering any effect in the inflammatory iCAF population.*

Response:

We thank the reviewer for this important comment. In the revised manuscript, in addition to myofibroblast subtype of cancer associated fibroblasts (myCAFs), we also examined the effect of PGRN on inflammatory CAFs (iCAFs) in the CKP tumors using mIF staining. By co-staining podoplanin (PDPN) and Ly6C as previously described by Steele *et al*, we found iCAFs (PDPN+Ly6C+) to be reduced in CKP tumors treated with PGRN antibody, although statistical significance was not reached (**Figure S5e**). By flow cytometry, we assessed the abundance of various CAF subtypes, including the relatively rare antigen-presenting CAFs (apCAFs), based on co-staining of PDPN, Ly6C and MHCII². Here, we observed again a slight decrease in iCAFs upon PGRN Ab (**Figure S5f**), although without statistical significance. However, PGRN Ab in general did not prominently change the composition as well as abundance of various CAFs (**Figure S5f**).

The corresponding new data is described in “Results” section (page 6,7, line 257-267).

Minor points:

→ *Figure 1a. PGRN staining is difficult to appreciate in the representative images, particularly in High grade PanIN and PDAC.*

Response: In the revised manuscript, images in **Figure 1a** are now shown at higher magnification.

→ *Was there any specific reason to detect MHC I in mice experiments recognizing only the H-2Db class I molecule?. Did the authors analyze effects on H-2Kb class I?*

Response: Both H-2Db and H-2Kb were in fact detected in the mouse tumors, and both of them showed similar trends upon PGRN modulation. However, since the neutralizing MHCI antibody we used in **Figure 7g-I** specifically neutralizes H-2Db, we only showed the results of H-2Db in the original version. In the revised manuscript, we have included the result of H-2Kb as well for better understanding of the effect of PGRN on MHCI in mouse settings (**Figure S7a, S8a**).

→ *Materials and Methods: There is no need to specify the method of euthanasia used in the study.*

Response: The corresponding description has been deleted in the revised manuscript.

Reviewer #2 (Remarks to the Author):

In this manuscript the authors unveiled a novel function of progranulin (PGRN) in the regulation of MHC I in PDAC. They demonstrate that tumor cell-derived PGRN drives MHC I downregulation in both murine tumors and human biopsies, correlating with lower CD8 infiltration. In addition, they demonstrate that PGRN blockade in vivo increases CD8 cytotoxicity and tumor infiltration. This effect is lost with simultaneous blockade of MHC I. The manuscript is well written, science is novel and methods and statistics are appropriate. This reviewer has the following comments:

→ 1. *While PGRN blockade had an effect on tumor progression in a genetic PDAC model, it did not in tumors expressing LCMV-gp33 in the absence of adoptive transfer of gp33-TCR Tg T cells. What are the possible reasons for such discrepancy?*

Response: We thank the reviewer for his review and helpful comment. We apologize for the understatement in the original manuscript when the sample size was too small (n=2-3) and quantification was not performed. In the revised manuscript, we were able to increase the number of mice in each control group to n=5. Although statistical significance was not reached, PGRN-induced effects indeed can be observed in the tumors expressing LCMV-gp33 in the absence of adoptive transfer of gp33-TCR Tg T cells, in terms of CD8 infiltration, granzyme B and cleaved caspase 3 levels, as well as tumor growth suppression (**Figure S9e,f**).

However, the effect of PGRN blockade is still not as profound as the one with adoptive transfer of gp33-TCR Tg T cells. This is because once encountering pathogen (LCMV-gp33), pathogen-specific T cells (LCMV-gp33-TCR Tg T cells) demonstrated higher expansion ability when compared to the recipient T cells³. The presence of LCMV-gp33-TCR Tg T cells is crucial to mount a profound anti-tumor cytotoxicity in the xenograft model, even though surface MHCI molecules are similarly increased upon PGRN Ab to present the tumor-specific antigens.

The corresponding new data is described in “Results” section (page 9, line 376-386).

→ 2. *Does PGRN blockade have any effect on long-term survival in these models?*

Response: Yes, we have performed long-term *in vivo* PGRN blockade to observe the effect on the survival of spontaneous PDAC mouse model in combination with chemotherapy in a separate study. PGRN Ab alone significantly prolonged the survival of the animals (**Figure R1**), but the effect of was not tremendous and the tumors were not cured completely. However, the result is indeed expected because although anti-tumor cytotoxicity is induced by increased surface MHCI upon PGRN blockade, PDAC is known to express relatively low level of tumor antigen due to low mutational load. With presumably low level of tumor antigens, effective and persistent anti-tumor cytotoxicity is likely not elicited. Therefore, we speculate that a combination of PGRN blockade with recently introduced therapies such as antigenic tumor

peptides or dendritic cells loaded with shared peptides might significantly augment the anti-tumor response and lead to long-term survival benefit. Such combination therapeutic regime, however, would be a separate study for the next stage, and beyond the scope of this manuscript.

→ 3. To really establish CD8 TILs as the main effectors upon PGRN blockade, authors should provide survival and/or antitumor data with CD8 depletion in a PDAC model.

Response: Thank you for the suggestion. In the revised manuscript, we performed the *in vivo* PGRN blockade in CKP mice with systemic CD8 depletion with antibody to demonstrate that CD8 cells are the main effectors of anti-tumor cytotoxicity upon PGRN blockade (Figure 5d-i). Upon CD8 depletion, the reduction of tumor burden induced by PGRN Ab was greatly abolished (Figure 5e,f). Depletion of CD8 cells was confirmed by flow cytometry (Figure 5g). IHC stainings showed that the PGRN Ab-induced reduction of PanCK+ tissues was restored by co-treatment of CD8 depletion Ab, while increased tumor infiltration of CD8+ and granzyme B+ cells were prominently abrogated (Figure 5h). As expected, cytotoxicity as reflected by cleaved caspase 3 staining in the tumors was also not observed upon CD8 depletion (Figure 5i). All the above findings strongly support that the anti-tumor cytotoxicity induced by PGRN blockade is mediated largely by CD8 cells.

The corresponding new data is described in “Results” section (page 7, line 288-297).

Reviewer #3 (Remarks to the Author):

The study by Cheung and colleagues describe a role for tumor cell expressed progranulin in regulation of MHC-I and immune evasion in pancreatic ductal adenocarcinoma. The authors show that increased PGRN expression in tumor epithelial cells increases with stage of malignancy and correlates with poor patient prognosis. Interestingly, macrophage-associated PGRN does not correlate with patient outcomes. Further, PGRN expression levels were shown to correlate with MHC-I expression and T cell infiltration in PDAC tumors – where PGRN low cells show higher MHC-I expression and increased T cell association. Accordingly, knockdown of PGRN or treatment with an PGRN blocking antibody is able to increase MHC-I levels and induce greater activated CD8+ T cell infiltration in FKP mouse model of PDAC. Mechanistically, the authors provide some data to suggest that PGRN suppression inhibits autophagy – which was recently shown to regulate MHC-I levels and surface expression in PDAC.

Finally the authors develop and utilize a new model antigen mouse model to further test the efficacy of utilizing an anti-PGRN ab for enhancing MHC-I expression, increasing T cell infiltration and blocking tumor growth. While a significant change in tumor growth was not observed in this model, increase MHC-I and T cell association as seen in the FKP studies were recapitulated. Overall this is a rigorous and well executed study that highlights a previously unrecognized role for tumor cell autonomous PGRN in modulation of immune evasion in PDAC.

→ However, the lack of clear mechanism connecting PGRN and MHC-I is a weakness of the study. Moreover, the claim that tumor cell autonomous effects exclusively effect MHC-I and immune evasion is not entirely justified given that ECM, stromal cell, endothelial cell derived PGRN are not considered. Therefore, my suggestions are mostly focused on strengthening these aspects of the manuscript.

Response: We appreciate the reviewer for the very constructive advice. Regarding the connection between PGRN and MHC-I expression, we have included a number of new experiments in the revised manuscript to illustrate the regulatory role of PGRN in lysosomal activity and autophagic flux (**Figure 3, S3, S4**), providing a potential mechanism underlying MHC-I regulation by PGRN.

The corresponding new data is described in “Results” section (page 5-6, line 166-216).

While for the role of stromal cell-derived PGRN, we totally agree and appreciate the importance of PGRN derived from the stromal cells, particularly from macrophages. Since PGRN is expressed in both tumor cells and macrophages, where different biological effects are exerted, our Ab approach is not able to distinguish the functions and significance of PGRN in the two compartments. Indeed, in addition to MHC-I regulation, we anticipate that PGRN also plays crucial role in other immune evasion mechanisms. In our study, we showed that PGRN blockade induced macrophage polarization from M2 to M1 phenotype (**Figure 4h**), which echoes the previous findings reported in metastatic PDAC in liver⁴. Besides, despite statistical insignificance, a slight decrease in iCAFs was also observed upon PGRN suppression (**Figure S5e,f**).

We observed in our xenograft model in nude mice that tumor initiation and growth of *GRN*-suppressed MiaPaCa2 cells was reduced when compared to control cells (**Figure R2a**). Tumor-promoting M2 macrophages were reduced, and inflammatory cancer-associated fibroblasts (iCAFs) also slightly decreased in *GRN*-suppressed MiaPaCa2 xenografts. However, anti-tumor cytotoxicity was not induced as indicated by unchanged cleaved caspase 3 expression (**Figure R2b,c**). These findings suggest that PGRN blockade not only induces CD8-mediated anti-tumor cytotoxicity, but also suppresses tumorigenesis through regulating other stromal components in the TME. Further investigations are required to comprehensively delineate the signaling pathways underlying PGRN-mediated immunoevasion in tumor cells.

In the revised manuscript, we have also discussed the importance of the impact of PGRN on stromal cells in the “Discussion” section (page 10, line 427-436). In addition, the data showing PGRN Ab-induced macrophage polarization is now shown as main figure (**Figure 4h**), instead of supplementary material, in the revised version. For the effect of PGRN on another important stromal component, cancer-associated fibroblast, CAFs, in addition to myofibroblast (myCAF), we have extended our investigation to other subtypes of CAFs, including inflammatory CAFs (iCAF) and antigen-presenting CAFs (apCAF) (**Figure S5d-f**). However, since the effect of PGRN blockade on the CAFs was not significant, the data remain to be shown as supplementary data.

The corresponding data is described in “Results” section on page 6-7, line 253-267.

→ 1. *If the authors wish to establish PGRN as an upstream regulator of autophagy in PDAC they should complement the data shown in figure 3 with autophagy flux assays which are standard in the field. These include use of the dual fluorescent LC3 autophagy reporter or western blot analysis of LC3 lipidation in the presence/absence of lysosome blockade in the PRGN WT/KD setting.*

Response: Thank you for the suggestion. We agree that although in the previous

version, we have shown an increase in autophagosomes in PDAC cells upon GRN suppression in terms of LC3B puncta size, it is still unclear whether the increased amount of autophagosomes is due to increased autophagosome synthesis or impaired clearance of autophagosomes.

Therefore, in the revised manuscript, we assessed LC3-II levels in the cells treated with or without V-ATPase inhibitor Bafilomycin A (BafA), which blocks the degradation of autophagosomes. If autophagy is induced upon GRN suppression, BafA treatment will increase the LC3B-II level; if clearance of autophagosomes is blocked, LC3B-II level will not be affected in the presence of BafA. Here, BafA increased LC3B-II in control cells (nc), but not the GRN-suppressed PDAC cells, which already showed augmented LC3B-II level when compared to controls (**Figure 3e, S3e**). The phenomenon was further confirmed by measuring the amount of p62/sequestosome-1 (SQSTM1), which has been implicated in autophagic cargo recognition and is lost in the late stage of autophagy during autolysosome degradation⁵. An increase in the amount of p62/SQSTM1 is related to the inhibition of autophagy flux. Immunoblotting results revealed that p62/SQSTM1 levels increased in GRN-suppressed cells (**Figure 3e, S3e**). This indicates that GRN suppression induced accumulation of autophagosomes, reflecting an inhibition of their degradation in PDAC cells. As expected, the increase in p62 was also observed in control cells treated with BafA, but not on GRN-suppressed cells (**Figure 3e, S3e**). The new findings therefore support our statement that PGRN regulates autophagic flux in PDAC.

Finally, to further confirm the effect of PGRN, we treated a low PGRN-expressing PDAC cell line with high surface MHC1 expression, HupT4 (**Figure S2a,d**), with recombinant PGRN (rPGRN). Upon treatment with rPGRN, surface MHC1 level of HupT4 cells decreased (**Figure S4a**). IF staining showed that the clear membranous MHC1 staining in HupT4 cells was greatly reduced upon rPGRN treatment (**Figure S4b**). LC3B puncta size was increased upon rPGRN treatment (**Figure S4b,c**). LC3B-II level in the cells was assessed by immunoblotting. Notably, rPGRN increased LC3B-II level in HupT4 cells (**Figure S4d**), which was further augmented upon co-treatment with BafA (**Figure S4e**). Taken together, our data demonstrated that an important role of PGRN in autophagy.

The corresponding new data is described in “Results” section (page 5, line 166-185; page5-6, line 208-216).

→ 2. Likewise, if the authors believe that PGRN loss impacts lysosome function then appropriate assays to test lysosome activity should be included (pH measurements, ability to degrade cargo, diameter measurements etc). For example, GRN^{-/-} microglia show lysosomal defects (PMC4860138). It would be interesting to know whether PGRN suppression in PDAC display similar defects.

Response: Based on the previous study on the regulatory of PGRN on lysosomal functions and biogenesis through acidification of lysosomes⁶, we speculated that PGRN loss should impact lysosome function. In the revised manuscript, we examined the effects of PGRN on lysosomal function. Because acidic pH is required for lysosomal activity⁷, we evaluated whether PGRN affected lysosomal pH by LysoSensor DND-189, which becomes more fluorescent in an acidic environment. GRN suppression induced a significant reduction in DND-189 signal when compared to control cells (**Figure 3h, S3h**), indicating an important role of PGRN in maintaining lysosomal acidification. Next, GRN-suppressed cells were assayed for their ability to process DQ-BSA (a derivative of BSA), which is quenched upon cleavage by proteolytic enzymes in lysosomes²². As revealed by IF staining and flow cytometry,

dequenching of DQ-BSA was significantly decreased in *GRN*-suppressed cells when compared to controls (**Figure 3i,j, S3i,j**). Upon treatment with BafA, dequenching of DQ-BSA was greatly diminished and there was no difference between *GRN*-suppressed cells and controls (**Figure 3i,j, S3i,j**). The above findings indicate the crucial role of PGRN in regulating proteolytic activity.

The corresponding new data is described in “Results” section (page 5-6, line 196-207).

→ 3. *It is unclear why the authors chose to conduct the shPGRN KD experiments in combination with serum starvation (Fig. 3c). Can the authors repeat these experiments in normal media conditions to definitely establish serum independent PGRN effects on autophagy?*

Response: We thank the reviewer for this important point. In the previous manuscript, we conducted the shPGRN KD experiments in combination with serum starvation because some previous studies showed that shPGRN KD experiments in combination with serum starvation^{8,9}. However, we also understand that this would make the data interpretation difficult as serum starvation induced autophagy. Therefore, in the revised manuscript, we have repeated all the shPGRN KO experiments in normal media conditions, i.e. 10% FBS (**Figure 3c-j, S3c-j**), so that PGRN effects on autophagy can be appreciated without the effect of serum starvation.

The corresponding data is described in “Results” section (page 5-6, line 166-207).

→ 4. *It is unclear the extent to which PGRN is secreted from PDAC cells versus intracellular localized and how this compares to macrophages or other cellular populations within the tumor microenvironment. Accordingly, is PGRN secretion important for its role in modulating MHC-I levels? Or is the function of PGRN in regulation of MHC-I restricted to an intracellular role?*

Response: PGRN is a growth factor that acts in autocrine manner¹⁰. Once being secreted, soluble PGRN binds to the PGRN-producing cells, and maintain the positive feedback of PGRN expression¹⁰. This is also supported by our finding that upon PGRN Ab treatment, both cellular and secreted PGRN levels of human PDAC cells MiaPaCa2 and Patu8988T were significantly reduced (**Figure S5a**). However, in the tumors, it is difficult to distinguish and compare the amount of secreted versus intracellularly localized PGRN in the tumors due to the dynamic process. Besides, since PGRN is expressed by both tumor cells and macrophages in primary PDAC, it is practically challenging to dissect the source of secreted PGRN in the *in vivo* setting.

Regarding the role of intracellular or secreted PGRN in MHC-I regulation, according to previous study performed in microglial cells and cervical cancer cells, secreted PGRN is incorporated into cells via sortilin or cation-independent mannose-6-phosphate receptor, and then facilitated the acidification of lysosomes and degradation of autophagosomes⁶. Therefore, regardless of the source of PGRN, when secreted PGRN is incorporated into tumor cells, lysosome activity and autophagic flux can be promoted and probably facilitate the degradation of MHC-I. This hypothesis is also supported by our findings that, both *GRN* knockdown (which represents reduced tumor endogenous PGRN level) (**Figure 3, S3**) and recombinant PGRN treatment (which represents exogenous PGRN of any source) (**Figure S4**), could demonstrate the role of PGRN in regulating MHC-I expression and autophagic flux in PDAC cells.

However, it is interesting to note that, while we assumed PGRN acts in autocrine manner on both tumor cells and macrophages, upon PGRN blockade in the spontaneous PDAC mouse model, MHC-I upregulation is mostly observed in tumor cells, but not in stromal cells, as assessed by the mIF staining and quantification (**Figure R3**). This suggests that PGRN might activate different pathways and mechanisms in tumor cells and macrophages. While further investigation is needed to clarify the different roles of PGRN in the two cell types, we would like to focus in this study on the role of PGRN in promoting immune evasion in PDAC by downregulating MHC-I via lysosomal and autophagic pathways in tumor cells.

→ 5. *The authors show in Fig. S2a that human PDAC cells show differential PGRN expression levels. How does this correlate with MHC-I levels and localization given that MHC-I is low across PDAC?*

Response: Although MHC-I levels are generally low in PDAC, there is heterogeneity in MHC-I levels and localization across different PDAC cell lines. In the revised manuscript, we included the flow cytometric data of surface and intracellular MHC-I levels in 8 human PDAC cell lines with differential PGRN expression (**Figure S2d**). We observed a negative correlation between PGRN and surface MHC-I levels. High PGRN-expressing cell lines such as MiaPaCa2 and Patu8988T showed lower surface but higher intracellular MHC-I expression; while the opposite was observed in low PGRN-expressing cell lines such as HupT4 and HPAC. Microscopic images further confirmed the prominent membranous MHC-I expression in HupT4 cells, a low PGRN-expressing cell line (**Figure S4b**). Although we believe there may be other mechanisms for the higher surface MHC-I levels in the low PGRN-expressing cell lines, the correlation between surface MHC-I and PGRN expression in the 8 human PDAC cell lines at least concur with our proposed regulatory role of PGRN in surface MHC-I in PDAC cells.

The corresponding new data is described in “Results” section (page 4, line 153-156; page 5-6, line 208-216).

→ 6. *It would be beneficial to provide quantification of the level of PGRN at different stages of PDAC progression (Fig.4a) in order to justify the statement that “as lesions progressed to early PDAC, PGRN levels reached the maximum and slightly decreased in advanced stage” – line 197.*

Response: In the revised manuscript, we have replaced the original IHC staining of PGRN with multiplex IF images with PGRN co-stained with tumor marker PanCK and M2 marker MRC1 (**Figure 4a**). Quantification of PGRN+ tumor cells (PanCK+) performed by software HALO was shown to support our statement regarding the trend of PGRN expression during PDAC development.

The corresponding new data is described in “Results” section (page 6, line 227-234).

→ 7. *What is the overall survival of FKP mice with or without PGRN antibody treatment? This information would be very useful to include.*

Response: Due to the fast-growing GP82 cells engrafted in the FKP mice, the animals had to be killed at second week after treatment started, as the tumor volumes already reached the ethical requirement, and therefore survival of FKP mice engrafted with GP82 upon PGRN Ab treatment cannot be assessed.

However, we have performed long-term *in vivo* PGRN blockade to observe the effect on the survival of spontaneous PDAC mouse model (*CKP*) in combination with chemotherapy in a separate study. Notably, PDAC development of *FKP mouse model* is similar to the *CKP* model, and the survival of the two models are virtually identical as reported previously^{11, 12}. PGRN Ab alone significantly prolonged the survival of *CKP* mice (**Figure R1**), but the effect of was not tremendous and the tumors were not cured completely. However, the result is indeed expected because although anti-tumor cytotoxicity is induced by increased surface MHC1 upon PGRN blockade, PDAC is known to express relatively low level of tumor antigen due to low mutational load. With presumably low level of tumor antigens, effective and persistent anti-tumor cytotoxicity cannot be elicited. Therefore, we speculate that combination PGRN blockade with recently introduced therapies such as antigenic tumor peptides or dendritic cells loaded with shared peptides might significantly augment the anti-tumor response and lead to long-term survival benefit. Such combination therapeutic regime, however, would be a separate study for the next stage, and no finding is available at this moment.

→ 8. *In figure 5 it is difficult to assess whether the increase levels of activation marks are simply reflective of increased infiltration of CD8+ T cells. It would be useful to establish % co-staining between CD8 and the activation markers to gain a clearer indication of the percentage of total CD8 cells that are also positive for granzyme B, Eomes, T-bet. This would justify the statement “PGRN Ab enhanced the cytotoxicity of the infiltrating CD8+ T cells”*

Response: Thank you for the suggestion. In the revised manuscript, multiplex IF stainings were performed to co-stain CD8 with cytotoxic marker granzyme B and T-bet in the PGRN Ab-treated tumors (**Figure 5b**). Quantification was performed by software HALO and we showed that a significant proportion of infiltrating CD8 cells was also positive for granzyme B or T-bet (**Figure 5b**). Eomes was not included in the multiplex staining because the eomes+ cells were relatively rare (**Figure S6b**), and the subsequent quantification of its co-expression with CD8 might not be reliable. However, with the new data of CD8 co-expression with granzyme B and T-bet, we think the statement “PGRN Ab enhanced the cytotoxicity of the infiltrating CD8+ T cells” is justified.

The corresponding new data is described in “Results” section (page 7, line 278-281).

→ 9. *Can the authors confirm via flow cytometry that anti-PGRN (KD or antibody treatment) leads to increased cell surface MHC-I in vitro or better yet in vivo?*

Response: In the original manuscript, flow cytometry was used to assess the cell surface MHC1 in human PDAC cell lines MiaPaCa2 and Patu8988T upon *GRN* suppression (**Figure 3a and S3a**). In the revised version, we included the flow cytometric data on surface MHC1 expression upon PGRN Ab treatment in the same cell lines (**Figure S7b**). While for mouse tumor cells, we showed in the original manuscript by flow cytometry that PGRN Ab significantly upregulated surface MHC1 expression on GP82 cells (**Figure 7e, S8a**). In the revised version, we measured the surface MHC1 expression on mouse tumor cells freshly isolated from spontaneous PDAC tumors treated with or without PGRN Ab (**Figure S7a**).

The corresponding new data is described in “Results” section (page 7, line 303-307).

→ 10. It is confusing as to why gp33 expressing tumors treated with PGRN Ab but without LCMV-gp33 reactive T cell injection do not show significant native CD8 T cell infiltration (line 319-322). The authors show that the tumor cells elevate MHC-I (Fig S7d), which should be sufficient to elicit increased native T cell infiltration against endogenous antigen as in the FKP model, and lead to some level of anti-tumor response. Could the authors explain why this does not occur? Overall the data generated utilizing the model antigen system, while useful, is less compelling than the results obtained using the FKP model. This is also reflected in the absence of a significant decrease in overall tumor volume T/Tu/GP/Pab arm.

Response: As described earlier in the response to reviewer #2, we apologize for the understatement in the original manuscript when the sample size was too small (n=2-3) and quantification was not performed. In the revised manuscript, we increased the number of mice in each control group to n=5. Although statistical significance is not reached, PGRN-induced effects indeed can be observed in the tumors expressing LCMV-gp33 in the absence of adoptive transfer of gp33-TCR Tg T cells, in terms of CD8 infiltration, granzyme B and cleaved caspase 3 levels, as well as tumor growth suppression (**Figure S9e,f**).

Yet, the effect of PGRN blockade is still not as profound as the one with adoptive transfer of gp33-TCR Tg T cells. This is because once encountering pathogen (LCMV-gp33), pathogen-specific T cells (LCMV-gp33-TCR Tg T cells) demonstrated higher expansion ability when compared to the recipient T cells³. The presence of LCMV-gp33-TCR Tg T cells is crucial to mount a profound anti-tumor cytotoxicity in the xenograft model, even surface MHC-I molecules are similarly increased upon PGRN Ab to present the tumor-specific antigens.

The corresponding new data is described in “Results” section (page 9, line 376-386).

Minor comments:

→ Clear definition of abbreviations for Figure 7g and H should be indicated in the legend. Statistical testing should be calculated between the T/Tu/GP vs T/Tu/GP/P-AB treatment arms in order to establish whether blocking PGRN can elevate T cell activation and anti-tumor cytotoxicity.

Response: Thank you for the comment. The legend of **Figure 7g** and **7h**, as well as the statistical test have been revised accordingly.

The corresponding amendment can be found on page 23, line 1013-1015).

→ In Fig. 2c MHC-I low status in PGRN negative section correlates to areas in which PanCK staining is also low. Is this representative and if so can the authors explain this?

Response: In Fig. 2c, the percentages of PanCK+ staining in the PGRN- and PGRN+ regions are indeed comparable as quantified by software. Most importantly, when performing quantification of the mIF staining (**Figure 2d**), percentage of MHC-I, as well as the number of CD8+GzmB+ cells, were all calculated based on panCK+ tumor cells. Besides, the analysis was performed on the whole tissue basis, i.e. all regions were included for quantification, and the images shown in **Figure 2c** are only for visualization to demonstrate the spatial interaction of PGRN, MHC-I and CD8 cells. To avoid the confusion, we have included “PanCK+” in the labels of **Figure 2d**.

→ Line 331 of the discussion relating to “antigenic tumor peptides or dendritic cells loaded with shared peptides” requires references.

Response: Corresponding references were added in the revised manuscript (page 9, line 396).

→ Line 334 of the discussion which states that PRGN in tumor cells is “a key instructive regulator of immune evasion” is overstated given that the exact mechanism as to how this occurs is not established and the data showing a block in autophagy is not convincing. This sentence should be revised if additional data in support of a role for PGRN in regulation of autophagy in PDAC is not provided.

Response: Given the new data supporting the role of PGRN in regulating lysosomal activity and autophagy in PDAC, we think this statement is justified in the revised version.

Reference:

1. Maurer, C. *et al.* Experimental microdissection enables functional harmonisation of pancreatic cancer subtypes. *Gut* **68**, 1034-1043 (2019).
2. Steele, N.G. *et al.* Inhibition of Hedgehog Signaling Alters Fibroblast Composition in Pancreatic Cancer. *Clin Cancer Res* **27**, 2023-2037 (2021).
3. Utzschneider, D.T. *et al.* T cells maintain an exhausted phenotype after antigen withdrawal and population reexpansion. *Nat Immunol* **14**, 603-610 (2013).
4. Nielsen, S.R. *et al.* Macrophage-secreted granulins support pancreatic cancer metastasis by inducing liver fibrosis. *Nat Cell Biol* **18**, 549-560 (2016).
5. Mizushima, N., Yoshimori, T. & Levine, B. Methods in mammalian autophagy research. *Cell* **140**, 313-326 (2010).
6. Tanaka, Y. *et al.* Progranulin regulates lysosomal function and biogenesis through acidification of lysosomes. *Hum Mol Genet* **26**, 969-988 (2017).
7. Kawai, A., Uchiyama, H., Takano, S., Nakamura, N. & Ohkuma, S. Autophagosome-lysosome fusion depends on the pH in acidic compartments in CHO cells. *Autophagy* **3**, 154-157 (2007).
8. Chang, M.C. *et al.* Progranulin deficiency causes impairment of autophagy and TDP-43 accumulation. *J Exp Med* **214**, 2611-2628 (2017).
9. Beel, S. *et al.* Progranulin functions as a cathepsin D chaperone to stimulate axonal outgrowth in vivo. *Hum Mol Genet* **26**, 2850-2863 (2017).
10. Zhou, J., Gao, G., Crabb, J.W. & Serrero, G. Purification of an autocrine growth factor homologous with mouse epithelin precursor from a highly tumorigenic cell line. *J Biol Chem* **268**, 10863-10869 (1993).
11. Cheung, P.F. *et al.* Notch-induced myeloid reprogramming in spontaneous pancreatic ductal adenocarcinoma by dual genetic targeting. *Cancer Res* (2018).
12. Schonhuber, N. *et al.* A next-generation dual-recombinase system for time- and host-specific targeting of pancreatic cancer. *Nat Med* **20**, 1340-1347 (2014).

Figure R1

Median survival (Post-treatment):
mIg (n=7): 7d
PGRN Ab (n=7): 13d
Log rank: $r = 6.134$, $p = 0.0133$

Figure R2

[Redacted]

Figure R3

1 **Reviewer Figure 1**

2 Kaplan-Meier overall survival plot showing significant survival benefit in CKP mice treated
3 with PGRN Ab (50mg/kg, n=7)) when compared to mIg controls (50mg/kg, n=7).

4

5 **Reviewer Figure 2**

6

7

8

[Redacted]

9

10

11

12

13 **Reviewer Figure 3**

14 Quantification of MHC+ (%) in tumor (PanCK+) and stromal (PanCK-) cells in CKP tumors
15 treated with mIg or PGRN Ab (PAb) (50mg/kg) for 2 weeks. n=5. **p<0.01

16

17

REVIEWERS' COMMENTS

Reviewer #1 (Remarks to the Author):

The authors have well-addressed all the comments.

Reviewer #2 (Remarks to the Author):

I am satisfied with the revisions made by the authors. All my concerns have been properly addressed.

Reviewer #3 (Remarks to the Author):

The authors have address the majority of my suggestions and generated a strong study in support of progranulin mediated regulation of autophagy and MHC-I levels in PDAC.

I have a few comments related to language and some inaccuracies in the text.

1. line 194-195: "the size of lysosomes increased...". Size was not measured here. Rather total Lamp1 fluorescence intensity was measured. This sentence should instead discuss "Lysosome number" or "Lysosome staining intensity.."

2. Line 202: The sentence "DQ-BSA is quenched upon cleavage by proteolytic enzymes in lysosomes" should read "DQ-BSA is dequenched upon cleavage by proteolytic enzymes in lysosomes" (leading to increased fluorescence). Therefore, the loss of DQ-BSA fluorescence in shPRGN cells is consistent with loss of lysosome proteolytic activity.

3. Regarding the claim in the abstract "we unveiled a cancer-cell autonomous function..." and line 400 of the discussion, I don't believe the authors have provided sufficient evidence in support of this claim. As noted in my prior comments, the contribution of ECM, stromal and endothelial cell derived PRGN is not addressed - (the authors respond to my point by focusing on macrophages and fibroblasts). I would suggest removing this statement from the abstract and focus instead on the finding that "Tumor derived but not macrophage derived PRGN" is important.

4. Given that the authors have largely confirmed a large body of recent literature indicating a critical role for autophagy/lysosome suppression in regulation of MHC-I / immune evasion (see references below) it may be prudent and scholarly to include these references and discuss the findings in the discussion.

PMC7190323; PMC7458662; PMID33443027; PMC7296553; PMC8205437; PMID32968282; Qiao Y et al Nature Cancer 2021; Poillet-Perez et al Nature Cancer 2020

5. The methods section for "Immuno-fluorescence staining" should be corrected as it currently includes use of Trametinib which is not used in this study.

REVIEWERS' COMMENTS

Reviewer #1 (Remarks to the Author):

The authors have well-addressed all the comments.

Response: Thank you for the positive comments and constructive suggestions to improve the manuscript.

Reviewer #2 (Remarks to the Author):

I am satisfied with the revisions made by the authors. All my concerns have been properly addressed.

Response: Thank you for the positive feedback, as well as the helpful comments and suggestions to improve our manuscript.

Reviewer #3 (Remarks to the Author):

The authors have address the majority of my suggestions and generated a strong study in support of progranulin mediated regulation of autophagy and MHC-I levels in PDAC.

I have a few comments related to language and some inaccuracies in the text.

1. line 194-195: "the size of lysosomes increased...". Size was not measured here. Rather total Lamp1 fluorescence intensity was measured. This sentence should instead discuss "Lysosome number" or "Lysosome staining intensity.."

Response: Thank you for the correction. We have changed the terms to "lysosome staining intensity" (line 197, highlighted).

2. Line 202: The sentence "DQ-BSA is quenched upon cleavage by proteolytic enzymes in lysosomes" should read "DQ-BSA is dequenched upon cleavage by proteolytic enzymes in lysosomes" (leading to increased fluorescence). Therefore, the loss of DQ-BSA fluorescence in shPRGN cells is consistent with loss of lysosome proteolytic activity.

Response: Thank you for the correction. We have made the correction accordingly (line 204, highlighted).

3. Regarding the claim in the abstract "we unveiled a cancer-cell autonomous function..." and line 400 of the discussion, I don't believe the authors have provided sufficient evidence in support of this claim. As noted in my prior comments, the contribution of ECM, stromal and endothelial cell derived PRGN is not addressed - (the authors respond to my point by focusing on macrophages and fibroblasts). I would suggest removing this statement from the abstract and focus instead on the finding that "Tumor derived but not macrophage derived PRGN" is important.

Response: We agree with the reviewer, and have replaced "cancer-cell autonomous function" with "the role of tumor-derived PGRN" in the abstract (line 51, highlighted).

4. Given that the authors have largely confirmed a large body of recent literature indicating a critical role for autophagy/lysosome suppression in regulation of MHC-I / immune evasion (see references below) it may be prudent and scholarly to include these references and discuss the findings in the discussion.

PMC7190323; PMC7458662; PMID33443027; PMC7296553; PMC8205437; PMID32968282; Qiao Y et al Nature Cancer 2021; Poillet-Perez et al Nature Cancer 2020

Response: Thank you for the suggestion. In the discussion section of revised manuscript, we have discussed our findings focusing on the role of autophagy/lysosome suppression in regulation of MHC-I / immune evasion, with the references kindly suggested by the reviewer (Line 406-416, highlighted).

5. The methods section for "Immuno-fluorescence staining" should be corrected as it currently includes use of Trametinib which is not used in this study.

Response: Thank you for the kind reminder. We have made the correction accordingly (Line 657-659, highlighted).